# T-cell trans-synaptic vesicles are distinct and carry greater effector content than constitutive extracellular vesicles

Pablo F. Céspedes [1,11✉], Ashwin Jainarayanan[1,11], Lola Fernández-Messina [2,3], Salvatore Valvo[1], David G. Saliba [1], Elke Kurz[1], Audun Kvalvaag[1], Lina Chen [1], Charity Ganskow [1], Huw Colin-York [1,4], Marco Fritzsche [1,4], Yanchun Peng [4,5], Tao Dong[4,5], Errin Johnson[6], Jesús A. Siller-Farfán [6], Omer Dushek [6], Erdinc Sezgin [7], Ben Peacock [8], Alice Law[8], Dimitri Aubert[8], Simon Engledow [9], Moustafa Attar [1,9], Svenja Hester[10], Roman Fischer [10], Francisco Sánchez-Madrid[2,3] & Michael L. Dustin [1✉]

The immunological synapse is a molecular hub that facilitates the delivery of three activation signals, namely antigen, costimulation/corepression and cytokines, from antigen-presenting cells (APC) to T cells. T cells release a fourth class of signaling entities, trans-synaptic vesicles (tSV), to mediate bidirectional communication. Here we present bead-supported lipid bilayers (BSLB) as versatile synthetic APCs to capture, characterize and advance the understanding of tSV biogenesis. Specifically, the integration of juxtacrine signals, such as CD40 and antigen, results in the adaptive tailoring and release of tSV, which differ in size, yields and immune receptor cargo compared with steadily released extracellular vesicles (EVs). Focusing on CD40L+ tSV as model effectors, we show that PD-L1 trans-presentation together with TSG101, ADAM10 and CD81 are key in determining CD40L vesicular release. Lastly, we find greater RNA-binding protein and microRNA content in tSV compared with EVs, supporting the specialized role of tSV as intercellular messengers.

[1] Kennedy Institute of Rheumatology, Nuffield Department of Orthopedics, Rheumatology and Musculoskeletal Sciences, The University of Oxford, Oxford, UK. [2] Immunology Service, Hospital de la Princesa, Instituto Investigación Sanitaria Princesa, Universidad Autónoma de Madrid, Madrid, Spain. [3] Intercellular communication in the inflammatory response. Vascular Physiology Area, Centro Nacional de Investigaciones Cardiovasculares (CNIC), Madrid, Spain. [4] MRC Human Immunology Unit, MRC Weatherall Institute of Molecular Medicine, Radcliffe Department of Medicine, The University of Oxford, Oxford, UK. [5] Chinese Academy of Medical Science (CAMS) Oxford Institute (COI), University of Oxford, Oxford, UK. [6] Sir William Dunn School of Pathology, The University of Oxford, Oxford, UK. [7] Science for Life Laboratory, Department of Women's and Children's Health, Karolinska Institutet, Stockholm, Sweden. [8] NanoFCM, MediCity, Nottingham, UK. [9] Oxford Genomics Centre, Wellcome Centre for Human Genetics, The University of Oxford, Oxford, UK. [10] Target Discovery Institute, Centre for Medicines Discovery, Nuffield Department of Medicine, The University of Oxford, Oxford, UK. [11] These authors contributed Equally: Pablo F. Céspedes, Ashwin Jainarayanan. ✉email: pablo.cespedes@kennedy.ox.ac.uk; michael.dustin@kennedy.ox.ac.uk

T lymphocytes are key players in the regulation and promotion of adaptive immunity. Helper T cells (TH), cytotoxic T lymphocytes (CTL), and regulatory T cells (Treg) shape cellular networks through the assembly of cell-cell interfaces termed Immunological Synapses (IS) with antigen-presenting cells (APCs) and other T cells[1,2]. These ISs enable three key receptor-ligand-driven signals critical to mounting an immune response or maintaining self-tolerance: (1) antigen recognition, (2) co-stimulation and co-repression, and (3) sensing of soluble cytokines released into the synaptic cleft. Here we propose a fourth type of trans-synaptic signal based on trans-synaptic supramolecular effectors and provide a general tool to study these challenging to isolate signaling entities.

In the past 10 years, several supramolecular effectors transferring information across the interface of cell-cell junctions have been identified. These include polarized exosomes (PE)[3], supramolecular attack particles (SMAPs)[4] released from multivesicular bodies into the synaptic cleft, synaptic ectosome (SE) budding from the T-cell plasma membrane across the synaptic cleft[5,6] and trans-endocytosis/trogocytosis of membrane fragments across the synaptic cleft[7–11]. PE and SE are extracellular vesicles actively formed by the donor T cells through the action of the endosomal sorting complexes for transport (ESCRT) machinery and together make up the trans-synaptic vesicles (tSV). SMAPs are proteinaceous particles similar in size to exosomes and SE but lack a phospholipid membrane and instead have a core-shell structure that enables the transfer of complex cargo[4].

The requirement for antigen sensing and the integration of multiple juxtacrine signals (i.e., trans-receptor-ligand interactions enabled by the nanometer proximity of the opposing cell membranes) enables a highly regulated and efficient exchange of supramolecular effectors across immunological synapses. In antigen-dependent contacts, T cells efficiently transfer extracellular vesicles containing T-cell antigen receptors (TCR) to APCs as early as 10–30 min of interaction[6,12]. Similarly, CD40 ligand (CD40L), which is essential for B-cell help and the initiation of germinal center reactions[13], is transferred from T cells to B cells in a contact-, CD40- and antigen-dependent manner[14,15]. The specialized molecular and organelle organization of the IS also enables the antigen-dependent delivery of microRNA (miR) to APCs[3], highlighting the critical role of the IS as a facilitator of a variety of intercellular messages. A more detailed characterization of trans-synaptic particles is nonetheless challenging due to the rapid internalization of trans-synaptic cargo by APCs and because cells engaged in contacts actively acquire membrane fragments via trans-endocytosis and trogocytosis[7–11], which altogether introduces confounding factors in the identification of T-cell-specific particulate effectors.

While tSV, trans-endocytosed fragments, and SMAPs are consumed by the synaptic partner, extracellular vesicles formed by the ESCRT machinery and constitutively released into the media (referred to herein as EVs) can be collected from the culture supernatant of activated T cells[16]. Previous work has used EVs as a surrogate for tSV. While this approach has had some predictive power, a comparison of tSV and EVs is critical for progress in the field of cell-cell communication.

Here, we develop a platform to specifically collect tSV released from three major types of T cells, study their regulation and compare their immune receptor, protein, and miR cargo with those of EVs. Our findings reveal unique regulatory and compositional features in tSV and support the notion that regulated cell-cell contacts, such as the immunological synapse, facilitate the exchange of particulate messengers. This platform should also be helpful for capturing tSV in other biological cell-cell junctions and in response to a variety of juxtacrine signals that can be modeled with supported lipid bilayers (SLBs).

## Results

**Trans-synaptic vesicles are larger and carry more cargo than EVs.** SLB on glass substrates have been used extensively to study cell recognition processes between live cells and SLB presents molecular components from a natural interaction partner. SLB on planar supports presenting surrogate antigen, adhesion, and co-stimulatory receptors were first used to characterize tSV and SMAPs from TH cells and cytotoxic T cells, respectively[4–6]. The advantage of the SLB is that following T-cell release of tSV[5,6] or SMAPs[4], the respective T cells can be selectively released, leaving the intact tSV or SMAPs behind for analysis by imaging or mass spectrometry. SLB can be readily formed on $5.0 \pm 0.05\,\mu m$ glass beads (BSLB) to improve compositional analysis[5]. We have further improved on this first generation of BSLB to develop a general resource for the study of tSV and SMAPs in any synaptic model.

Our second-generation BSLB model uses a surrogate antigen, a 14 His-tagged α-CD3ε Fab, that can be gently eluted from the SLB to enable the release of tSV. We used fluorescence correlation spectroscopy to demonstrate that the recombinant α-CD3ε Fab and a co-incorporated Abberior Star-Red phosphatidylethanolamine had diffusion coefficients of $0.68 \pm 0.39$ and $3.2 \pm 0.79\,\mu m^2/s$, respectively, which are comparable to ranges reported for PM proteins and lipids[17–19]. To analyze tSV from TH cells we used BSLB presenting a range of α-CD3ε Fab densities, and physiological densities of the adhesion molecule ICAM1 (200 molec/μm²) and costimulator receptor CD40 (20 molec/μm²), which are characteristic of natural APCs. Time-lapse imaging of T-cell-BSLB co-cultures demonstrated that BSLB instigated the transfer of CD40L from stimulated T cells, which left a "synaptic stamp" on the engaged BSLBs composed of the transferred tSV (Fig. 1a). To enable the synchronous release of BSLBs from conjugates, we gradually cooled the cell-BSLB co-culture to 4 °C to promote dissociation of the LFA-1-ICAM1 interaction while minimizing mechanical stress on BSLB and their tethered tSV. Following this treatment, flow cytometry (FCM) revealed the conversion of conjugates to single T cells and BSLB with evidence of TCR downregulation on T cells and TCR acquisition by BSLB as a function of α-CD3ε Fab density (Fig. 1b and Supplementary Fig. 1a), as previously reported[5]. While BSLB acquired little CD2 and CD4, T cells acquired little fluorescent lipids (DOPE) indicating limited internalization of BSLB material (Fig. 1b and Supplementary Fig. 1b), contrasting with what has previously been reported for B-cell IS[20].

We next eluted the tSV with ice-cold EDTA from BSLB previously enriched by low-speed centrifugation (100 g 1 min) and used differential and ultracentrifugation to concentrate the eluted tSV (see Methods). We focused our analyses on tSV containing the T-cell effector molecules TCR and CD40L, the broad EV and PM marker CD81+, and on CD63+ tSV belonging to exosomes. Most of the TCR, CD40L, and CD81 present on BSLBs were eluted under these conditions (Fig. 1c, d–f). Elution of CD63+ vesicles was less efficient but still significant (Fig. 1g). Electron microscopy images of eluted tSV revealed classical EV-like profiles (Fig. 1h, arrows). We were now able to compare tSV with steadily released EVs isolated from activated TH cells of the same donors cultured in EV-free media. TSV was larger than EVs by nanoparticle tracking analyses (NTA) (Supplementary Fig. 1c). This was confirmed by nanoFCM revealing tSV having a median diameter of $82.13 \pm 0.75$ nm to $84.4 \pm 5.99$ nm, with or without CD40 in BSLB, respectively, compared to EVs from T cells of the same donors with a median diameter of $65 \pm 25$ nm. More detailed analyses using silica bead standards for size segmentation revealed a higher frequency of events larger than 113 and 155 nm in tSV compared with EVs (Supplementary Fig. 1d). EVs showed a higher frequency of events in the 68 nm size bin

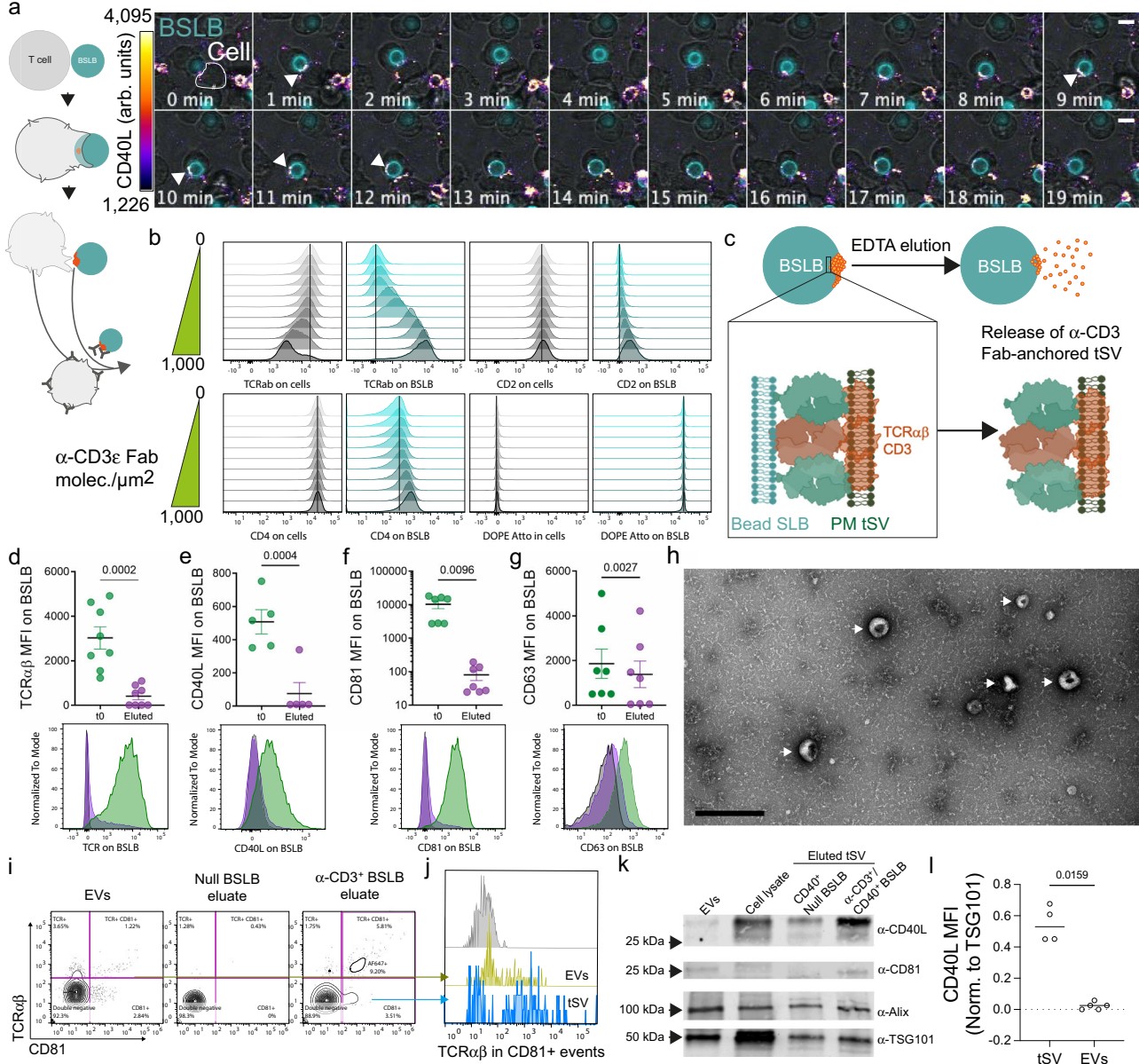

**Fig. 1 As synthetic APC, BSLBs trigger synapse formation and the release of tSV by stimulated T cells. a** Time-lapse confocal microscopy showing the interaction between TH and BSLB and the active transfer of CD40L$^+$ synaptic stamps to BSLB (white arrowheads). CD40L was tracked using 1 µg/mL of anti-CD40L clone 24–31. Scale bar = 5 µm; arb. units = arbitrary fluorescence units. **b** FCM analyses of vesicular transfer from TH (gray histograms) to BSLB (teal histograms) presenting 2-fold titrations of α-CD3ε Fab (0–1000 molec./µm$^2$) ICAM1 (200 molec./µm$^2$) and CD40 (20 molec./µm$^2$). Vertical lines indicate the median of cells and BSLB in the condition with no α-CD3 Fab (null). **c** After conjugate separation by cold, tSV are released from BSLBs with the use of 50 mM EDTA. **d–h** Elution results in the release of **d** TCR$^+$, **e** CD40L$^+$, **f** CD81$^+$, and to a lesser extent **g** CD63$^+$ tSV, from the surface of BSLB as measured by FCM staining comparing null BSLB (black histograms), with BSLBs prior to (green), and after elution (violet). **h** Transmitted electron microscopy of eluates reveals the presence of tSV (white arrows) and small soluble proteins. Scale bar = 500 nm. **i** Representative NanoFCM analyses showing the detection of TCRαβ$^+$ and CD81$^+$ events in isolated EVs (left panel), and the eluted material from Null BSLB (middle panel) and α-CD3$^+$ BSLB (tSV, right panel). **j** Overlaid histograms showing TCRαβ expression on CD81$^+$ vesicles from EVs (yellow) and tSV (light blue) compared to double negatives (gray). **k** Immunoblot for comparison of CD40L, CD81, Alix, and TSG101 in EVs and eluted tSV deriving from 5 × 10$^6$ of either cells or BSLB, respectively. Whole-cell lysate represents a total of 2.25 × 10$^5$ cells. The bands shown are from the same immunoblot membrane. **l** CD40L levels in eluted fractions (tSV) and EVs as measured by immunoblot (shown is CD40L normalized to TSG101). Normality was determined using Shapiro–Wilk test and statistical significance was determined by the two-tailed unpaired *t* test (**d–g**) and by the unpaired, two-tailed Mann–Whitney test (**l**). Data represents mean ± SEM of n = 2 (**a**, **i–j**) and n = 4 independent experiments (**b**, **e–h**, **k–l**). Uncropped and unprocessed versions of the immunoblots shown in **k** are available as source data.

(Supplementary Fig. 1d). A comparison of marker-specific vesicles within tSV and EVs revealed that CD81[+] tSV coexisted with TCRαβ[+] at higher frequencies in tSV than EVs (Fig. 1i–j) and a larger median size for TCR[+] tSV than TCR[+] EVs (Supplementary Fig. 1e–f), respectively. We also observed a larger median size for CD40L[+] or CD81[+] tSV than CD40L[+] or CD81[+] EVs (Supplementary Fig. 1e), whereas no significant size differences were observed for BST2[+] or CD63[+] tSV and EVs. TSV also expressed higher levels of TCR and CD40L than EVs (Fig. 1j–l, and Supplementary Fig. 1g–h). Since CD40L detection by antibodies is partly impaired by the presence of CD40[5], we confirmed CD40L elution and compared its expression levels in tSV and EVs by immunoblotting (Fig. 1k and Supplementary Fig. 1h). Most likely, the overall larger size and immune receptor content of tSV derives from the subpopulation of SE, which contains high densities of TCR and CD40L as microclusters >80 nm[5]. The membrane-anchored full-length CD40L was predominantly found (29.2 kDa) in the eluates of BSLBs, with negligible detection of its soluble ectodomain (Supplementary Fig. 1h). Null CD40[+] BSLB captures negligible amounts of CD40L from TCR-unstimulated cells, which might relate to the CD40-dependent capture of a fraction of EVs released in response to adhesion (i.e., ICAM1:LFA-1 interaction). The imaging of tSV markers within T-cell-SLB interfaces with TIRFM and confocal imaging further revealed that beyond differences in size and immune receptor content, tSV displayed different degrees of co-localization in intact, non-permeabilized ISs (Supplementary Fig. 2a–m) reflecting their different subcellular origin and the compositional and spatial heterogeneity of tSV in the synaptic cleft.

TSV is likely composed of TCRαβ[+], CD40L[+] SE, and CD63[+] PE, which may have different release characteristics. We used the BSLB system to investigate requirements for tSV release using a panel of 12 pharmacological inhibitors of candidate steps. Inhibitors of cytoskeleton dynamics and the transport of vesicular cargo from different organelles, such as multivesicular bodies (MVB) and the trans-Golgi network (TGN) were used (Supplementary Data 1). Following a pre-treatment for 30 min and then the co-culture of T cells and BSLBs for another 90 min at 37 °C and 5% $CO_2$ (in the presence of inhibitors), we monitored the cell surface expression of TCR, CD40L, BST2, and CD63 (Supplementary Fig. 3a–d), their synaptic stability (Supplementary Fig. 3e), and their normalized trans-synaptic transfer to BSLB as the percent of the maximum observed in untreated controls (% Tmax) (Supplementary Fig. 3f–i).

Actin cytoskeleton disruptors Latrunculin A and Jasplakinolide, but not Cytochalasin D (CytD), impaired cell-BSLB conjugates (Supplementary Fig. 3e). CytD and the other inhibitors didn't impair cell-BSLB conjugate formation and thus allowed interrogation of distinct mechanistic steps in the delivery of tSV cargo. Fortuitously, we observed a differential inhibition in the transfer of TCR[+], CD40L[+], BST2[+], and CD63[+] tSV from TH cells to BSLB, suggesting distinct mechanisms of cargo delivery. The transfer of TCR[+] tSV was selectively promoted by dynamin inhibition by Dynasore, and conversely, strongly reduced by inhibition of actin polymerization, ESCRT machinery, and neutral sphingomyelinases (Supplementary Fig. 3f). Reduced endocytosis or increased ubiquitination of TCR following Dynasore treatment[21] might contribute to the observed increase in TCR[+] tSV transfer. The transfer of CD40L[+] tSV was strongly reduced following acute inhibition of several types of machinery in the hierarchy TGN > actin dynamics and ESCRT > ceramide synthesis > dynamin > vacuolar $H^+$-ATPases, and class I and II PI3K (Supplementary Fig. 3g). Other members of the TNF superfamily, such as FasL, have also been reported to be affected by interference with TGN trafficking[22,23], suggesting a conserved

mechanism of TNFSF delivery via tSV. In contrast, the transfer of BST2[+] (Supplementary Fig. 3h) and CD63[+] (Supplementary Fig. 3i) tSV was affected by inhibiting endosomal and lysosomal transport but not by inhibiting TGN transport. CD40L > TCR > BST2 showed a higher sensitivity than CD63[+] tSV to the inhibition provided by the MG132-induced depletion of free-ubiquitin[5,24,25], suggesting that MVB-associated stores are not affected by the acute pharmacological inhibition of ESCRT-I. The significant reduction of CD40L[+] and CD63[+] tSV transfer following N-SMases inhibition likely results from interference with broader membrane trafficking events involving both the TGN and endosomes where N-SMases preferentially locate to regulate membrane curvature and budding[26,27]. Finally, compared to other vesicle subpopulations, the production of CD40L[+] tSV requires quick antigen-dependent upregulation of CD40L on the plasma membrane, suggesting that the majority of its trans-synaptic transfer occurs as part of budding synaptic ectosomes (Supplementary Fig. 4a–d). While BFA and manumycin inhibited the α-CD3-induced upregulation of CD40L, CytD impaired CD40L[+] tSV release despite its effective cell surface upregulation (Supplementary Fig. 4b).

**T-cell subsets have distinct tSV cargo.** Next, we sought to evaluate whether different human T cells, including CD8[+] CTL, CD4[+] T helper cells (TH), and cultures enriched in CD127[low]CD25[high] FoxP3[+] cells (Treg), displayed distinctive tSV transfer hallmarks. All cells were isolated from peripheral blood, activated, and further expanded ex vivo as detailed in Methods (see Supplementary Fig. 5a–c). We evaluated the transfer of the TCRαβ heterodimer, CD2, the co-receptors CD4 or CD8, CD28, CD45, the tSV proteins CD63, CD81, and BST2, and effectors including regulatory enzymes (CD38, CD39, and CD73), and helper (CD40L) and cytotoxic (Perforin) proteins (Fig. 2a–p). Analyses were focused on markers expressed specifically by these populations and the MFI of single T cells and single BSLB after gentle dissociation of conjugates (see Supplementary Fig. 1a for a representative gating strategy). A normalized synaptic transfer metric was defined as the percent enrichment of T-cell proteins on a single BSLB from the total mean signal observed in single BSLB and single cells (see Eq. (1) in Methods and ref. [5]).

Because we reconstituted our BSLBs in the absence of ligands for TCR co-receptors CD4/CD8 (i.e., pHLA) and CD2 (i.e., CD58), we therefore expected the vesicular shedding of TCR to include limited co-receptors and CD2 (Fig. 2b–c). Compared with TH, CTLs transferred significantly more TCR and coreceptor in the range of α-CD3ε-Fab densities tested and consistently showed smaller α-CD3ε-Fab $EC_{50}$ values of 251.2 for TCR (compared with 1,251 in TH) and 50.87 for the coreceptor (compared with 56.44 in TH) (Fig. 2a–b). The higher antigen sensitivity of CTL compared with TH and increased physical proximity between TCR and CD8[28], might promote their sorting in tSV. Similarly, using chimeric antigen-receptor expressing T cells (CART) specific for HLA-A*02: NY-ESO-1[157–165] we tested whether the T1 CAR displays similar transfer dynamics than TCR. Interestingly, we observed a higher transfer of the CAR than TCR at comparable densities of α-CD3ε-Fab and agonistic HLAp complexes (Supplementary Fig. 5d). CART also transferred higher amounts of CD8, suggesting that CAR triggering by agonist pHLA, even at low densities, promotes the efficient release of CAR and CD8 (Supplementary Fig. 5e). Like CD2 and co-receptors, and despite their high level of cell surface expression, we observed a lack of CD28 and CD45 enrichment in the tSV released by different T-cell subsets (Fig. 2d–e). While CD45 is normally excluded from the cSMAC and the synaptic cleft, the co-receptors, CD2 (at exceptionally low levels) and

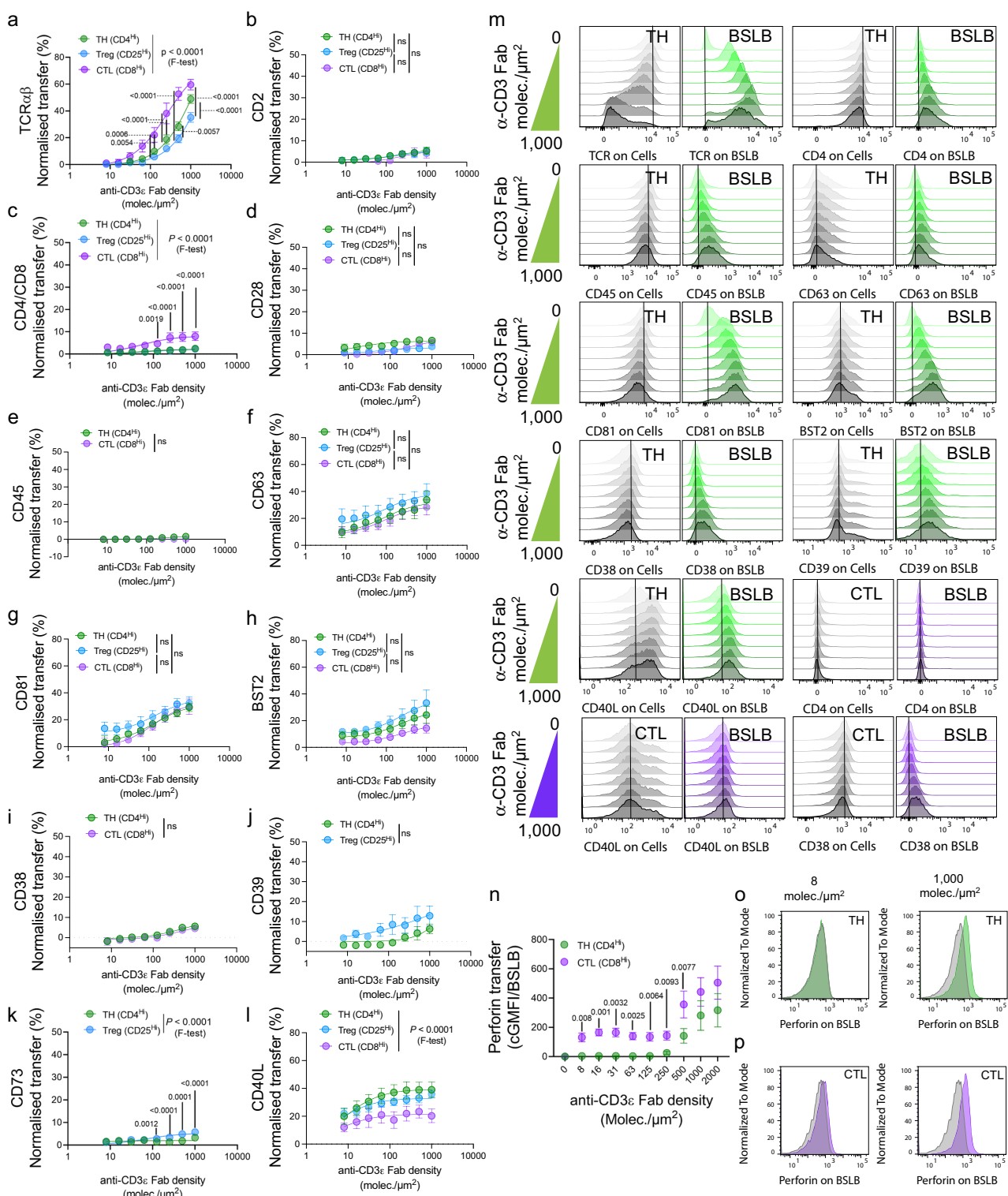

CD28 require binding to cognate ligands to be mobilized as microclusters to the synapse center[29–31]. The latter suggests that tSV heterogeneity is partly defined by APC ligands dynamically feeding information to T cells, which we explored further in the next section. The limited enrichment of CD4, CD8, and CD45 in tSV makes them suitable secondary parameters for discriminating single BSLBs in FCM analyses and sorting (Supplementary Figs. 1a, 5i and 9a).

TH, CTL, and Treg transferred comparable levels of CD63[+], CD81[+], and BST2[+] tSV to BSLB (Fig. 2f–h), indicating conserved

mechanisms delivering these components to the synaptic cleft. Consistently, CART showed a comparable transfer of CD63[+] tSV when stimulated through the CAR and the TCR (Supplementary Fig. 5f). We next measured the vesicular transfer of effector ectoenzymes, namely the ADP-ribosyl cyclase CD38 and the AMP- and adenosine-producing ectonucleotidases CD39 and CD73 (Fig. 2i–k). While CD38 polarizes to the IS[32] and participates in the generation of T-dependent humoral immunity[33], CD39 and CD73 are known tolerogenic mediators[34–36]. TH and CTL showed a limited, conserved, and

**Fig. 2 The synaptic transfer of vesicular effectors relates to the functional properties of different T-cell subsets.** Flow cytometry analyses measuring the transfer of various tSV to BSLBs presenting α-CD3ε Fab (0–1000 molec./μm$^2$), ICAM1 (200 molec./μm$^2$), and CD40 (20 molec./μm$^2$) were used in all experiments (see Supplementary Fig. 1a for a representative gating strategy). **a–p** Percent Normalized Synaptic Transfer (NST%) to BSLB from TH (green circles), Treg (light blue circles), and CTL (violet circles). Markers included **a** the antigen-receptor heterodimer TCRαβ; **b** CD2; **c** TCR co-receptors (CD4 or CD8); **d** CD28; **e** CD45; **f** CD63; **g** CD81; **h** BST2; **i** CD38; **j** CD39; **k** CD73, and **l** CD40L. **m** Representative histograms for most makers are shown on the right for TH (green histograms for BSLBs) and CTL (violet histograms for BSLBs). **n** Perforin transfer as null BSLB-corrected GMFI. **o–p** Representative histograms showing perforin deposition on BSLBs after co-culturing with TH (**o**) and CTL (**p**). Shown are overlaid histograms depicting BSLBs coated with different densities of anti-CD3 Fab. Data represent means ± SEM of n = 10 (**a**), 4 (**b, d**), 5 (**g, i, l**), and 6 (**c, g, f, h, j, k, n**), independent biological samples (donors) and experiments. Normality was determined using Shapiro–Wilk test and the statistical significance was determined by two-tailed Multiple *t* test with Holm-Šídák corrections (α = 0.05) for the multiple comparisons of tSV transfer across different α-CD3ε Fab densities and among different T-cell populations (**a–l, n**). Also, the comparison of α-CD3 Fab EC$_{50}$ and transfer maximum (Tmax) among T-cell populations was determined using *F* test fitted with three to four parameters (**a–l**). ns = not significant. Adjusted *P* values are shown next to each significant comparison.

comparable transfer of CD38$^+$ tSV to BSLB (Fig. 2i). Treg, on the other hand, showed an increased transfer of CD39 (Tmax = 17.83% compared with 6.5% in TH; Fig. 2j), and CD73 (Tmax = 5.73 % compared with 1.9% in TH; Fig. 2k), which associated with a higher central clustering of CD39 and CD73 in the cleft of Treg synapses (Supplementary Fig. 5j–o). The inclusion of itinerant enzymes such as CD38, CD39, and CD73 in tSV, although limited, might exert a feed-forward regulatory effect as shown for tumor-derived EV[37].

We also compared the transfer dynamics of CD40L and perforin, two major effectors mediating the activation and killing of APCs, respectively. Upon TCR-triggering, we observed a conserved relative transfer of CD40L in T cells, with TH secreting the most (Fig. 2l–m). Conversely, compared to TH, CTLs showed a reduced threshold for perforin release, which was transferred to BSLB at α-CD3ε-Fab densities as low as 8 molec./μm$^2$ for CTLs (Fig. 2n–p). In CART cells, perforin release followed a different behavior than exocytic CD63$^+$ tSV. More specifically, TCR-elicited perforin deposition on BSLB was higher than that resulting from CAR-triggering (Supplementary Fig. 5g), suggesting differential mechanisms influencing the release of tSV and cytotoxic SMAPs at the IS. Altogether, our data indicate that (1) T-cell subsets display differential dynamics of tSV transfer consistent with their effector functions and receptor-ligand interactions occurring at the IS and that (2) BSLB are excellent tools to study the biogenesis of both tSV and perforin-containing SMAPs.

**Juxtacrine signals and the cell's phenotype determine tSV cargo.** In the dynamic exchange of information between T cells and APCs several factors might influence the effector cargo of tSV. Hence, we harnessed BSLB to study the dynamics influencing CD40L$^+$ tSV release under near-physiological conditions and using absolute quantifications (see Methods). First, we measured the densities of CD40 on different human B-cell subsets (ranging between 16 and 646 molec./μm$^2$; Fig. 3a). Using these near-physiological estimates we then measured the transfer of CD40L$^+$ tSV to BSLB covering a range of 0 to 540 molec./μm$^2$ of CD40. As shown in Fig. 3b, we found that while freshly isolated and quiescent TH transferred negligible amounts of CD40L, activated and blasted TH showed high sensitivity to CD40, transferring CD40L to CD40 densities as low as 2 molec./μm$^2$. However, this high efficiency reached a maximum transfer (Tmax) at ~20 molec./μm$^2$ of CD40, which at higher densities outcompeted the anti-CD40L detection antibody[5] (Fig. 3b). The latter is expected as CD40 displays a high-affinity for its ligand (0.5–7.13 nM[38]). Therefore, we used the physiological minimum of CD40 (20 molec./μm$^2$) to study the dynamics affecting CD40L sorting in tSV. We used a vesicle stamp area of 0.82 μm$^2$ [25] to estimate and compare the effective densities of CD40L on the vesicular stamps of BSLB. Even after 4 h, no significant transfer of

vesicular CD40L was observed from quiescent cells (Fig. 3c). In contrast, activated TH blasts transferred significant amounts of CD40L and led to densities up to 1000 times those found on the plasma membrane, suggesting a significant gain in CD40L binding valency resulting from its vesicular packing. Remarkably, higher CD40L$^+$ tSV transfer in activated cells related to their higher expression of TSG101 (Fig. 3d–e), indicating that T-cell activation leads to the simultaneous upregulation of effectors and ESCRT-I. Further comparison of TH, Treg, and CTL revealed phenotype-specific differences in the dynamics of CD40L transfer among activated T cells, with the amounts released in the order TH > Treg>CTL (Fig. 3f; *F* test, *p* < 0.0001).

Next, to evaluate the effect of TCR sensitivity on CD40L tSV release we used 3 TH clones specific for the same influenza H3 hemagglutinin peptide HA$_{338–355}$ (NVPEKQTRGIFGAIAGFI) but displaying different TCR:pMHC potencies (Supplementary Data 2). T-cell clones showed comparable levels of CD40L transfer at higher antigenic HLA densities (Fig. 3g) and formed morphologically similar synapses (Fig. 3h–i). However, contrary to their IFN-γ response (Supplementary Data 2), clones showed differences in the HLA-DR9:HA EC$_{50}$ for CD40L transfer with clone 40 displaying the lowest (*P* = 0.0002, EC$_{50}$ = 22.02, compared to 66.01 and 101.9 for clones 16 and 35, respectively), suggesting that TCR:pMHC affinity has more substantial effects on the release of CD40L$^+$ tSV at lower antigen densities. The latter phenomenon might underpin the maintenance of an active CD40L-CD40 pathway in autoimmune diseases, where high densities of self-antigens and low affinities of self-reactive TCRs might suffice to trigger the assembly of non-canonical immune synapses[39] and the release of CD40L$^+$ tSV.

Beyond antigens (signal one), APCs present other juxtacrine signals (signal two) to fine-tune the activation of engaged T cells. To evaluate whether such signals shape the composition and release of tSV, we incorporated the co-stimulatory ligands CD80, CD86, ICOSL, the CD2 corolla-inducing ligand CD58[31], the TNF ligand superfamily (TNFSF) members OX40L (TNFSF4) and 4-1BBL (TNFSF9) and the co-repressors PD-L1 and PD-L2, whose interplay with the CD40-CD40L dyad remains poorly characterized. To evaluate whether viral proteins also shape tSV, we included the HIV-1 gp120 protein, which is a known high-affinity ligand for CD4[30,40–42]. We then compared the NST% for TCR (Fig. 4a), CD2 (Fig. 4b), CD28 (Fig. 4c), CD63 (Fig. 4d), CD81 (Fig. 4e), BST2 (Fig. 4f), CD40L (Fig. 4g–i) and CD4 (Fig. 4j–k) between different BSLB compositions using those containing only ICAM1, CD40, and increasing α-CD3ε-Fab densities as controls. While CD58 incorporation slightly promoted the transfer of CD2 (Fig. 4b), incorporation of CD80, CD86, and ICOSL induced an increased synaptic transfer of CD28 (Fig. 4c), and to a lesser extent that of CD40L (Fig. 4g). Most remarkably, PD-L1/PD-L2 inclusion on BSLB led to a significant reduction in the transfer of CD40L$^+$ tSV (Fig. 4g–i),

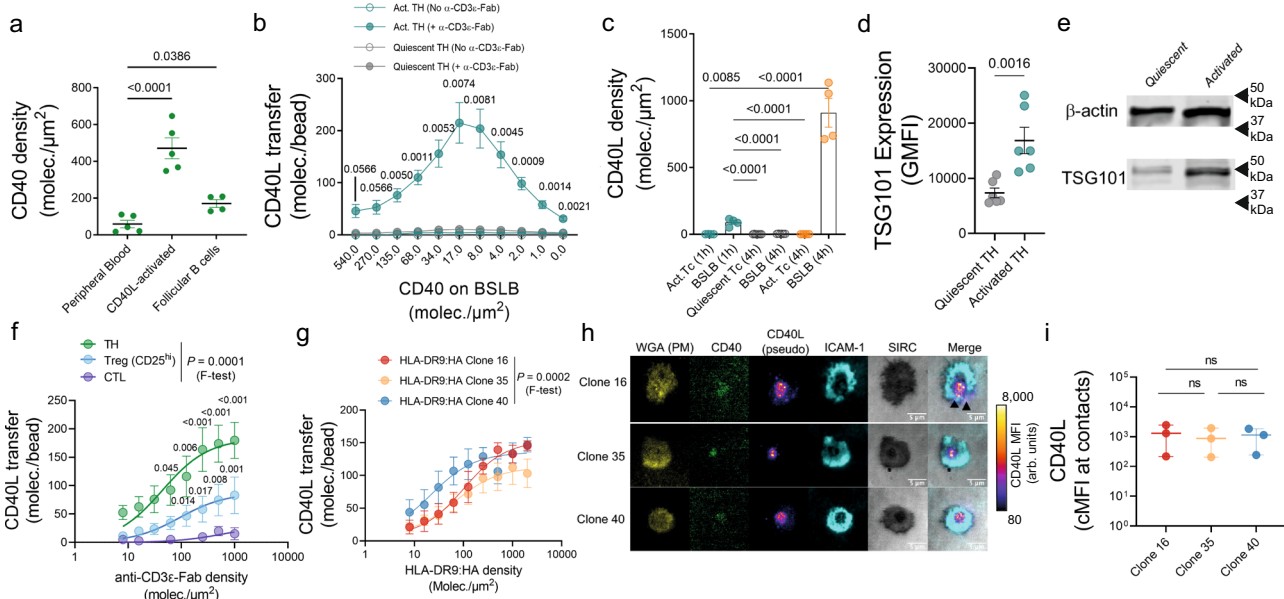

**Fig. 3 BSLBs unravel the dynamics influencing the biogenesis of CD40L⁺ tSV. a** CD40 densities on the surface of human B cells isolated from either peripheral blood (CD19hi HLA-DR⁺ and CD40⁺) or palatine tonsils (also CXCR5hi). **b** Transfer of CD40L⁺ tSV to BSLBs reconstituted with physiological densities of CD40 (as defined in **a**) and either no or 1000 molec./μm² of α-CD3ε Fab. Quiescent and α-CD3/CD28-activated and expanded TH are compared. **c** Density of CD40L molecules transferred to BSLB at different times by quiescent and activated cells. CD40L density was estimated using a previously defined[5] patch area of 0.82 μm². **d** TSG101 expression in quiescent and activated TH as measured by flow cytometry. **e** Immunoblots showing TSG101 expression in quiescent and activated TH. **f** Absolute CD40L transferred to BSLBs from TH, Treg, and CTL. **g** Absolute CD40L transferred to BSLBs from T-cell clones expressing TCRs with different potencies for the antigenic HLA-DRB1*09:01:HA$_{338-355}$ complex. **h** TIRFM imaging of CD40L clustering within synapses of T-cell clones stimulated on SLB containing 30 molec./μm² of antigen, 200 molec./μm² of ICAM1, and 20 molec./μm² of CD40. Scale bar = 5 μm. MFI = mean fluorescence intensity, arb. units = arbitrary fluorescence units. SIRC = surface interference reflection contrast; Plasma membrane (PM) glycans are labeled with Wheat Germ Agglutinin (WGA). **i** Measurement of CD40L interfacial clustering in the immune synapses of clones shown in **h**. Data represents means ± SEM of n = 4 (**a**, **c**), 6 (**b**, **d**), and 3 (**f–i**) biologically independent samples (donors) across 4 (**a**, **c**), 2 (**b**), and 3 independent experiments (**d–i**). Please refer to Supplementary Fig. 1a for a representative gating strategy showing the identification of single BSLBs and T cells by flow cytometry. Normality was determined by Shapiro–Wilk test and Statistical significance was determined by One-Way ANOVA corrected for multiple comparisons using FDR (Q = 0.05) and the two-stage step-up method of Benjamini, Krieger, and Yekuteli (**a**, **c**), two-tailed Multiple t test with Holm–Šídák test correction (α = 0.05) for the multiple comparisons of CD40L⁺ tSV transfer between activated and quiescent TH (**b**), and two-tailed paired T test (**d**). α-CD3ε-Fab EC50 and maximum transfer (Tmax) were calculated using three to four parameters F test (**f**, **g**). Adjusted P values are shown next to each significant comparison. Uncropped and unprocessed versions of the immunoblots shown in (**e**) are available as source data.

and a less significant reduction of TCR⁺, CD28⁺, CD63⁺, CD81⁺, and BST2⁺ tSV (Fig. 4a–f). Similarly, 4-1BBL/OX40L trans-presentation impaired TCR, CD81, CD40L, and BST2 transfer to BSLBs (Fig. 4a,e–i). Importantly, even in the absence of TCR stimulation, gp120 promoted CD4 transfer to BSLB (Fig. 4j–k), which might result from both the physical engagement of CD4 and the TCR-independent phosphorylation of Src, Lck, and CD3ζ triggered by gp120[30]. The gp120-induced transfer of CD4 and the CD80/86-instigated transfer of CD28 thus demonstrate that cognate and non-native juxtacrine signals shape the composition of tSV. These data suggest that the immunological synapse facilitates the delivery of specific, targeted, and adaptive trans-synaptic messages contingent upon ligands bound at the cell-cell interface.

Next, we used CRISPR/Cas9 genome editing to test the functional relevance of endogenous elements participating in tSV biogenesis and the synaptic release of CD40L. Because we used α-CD3ε Fab for the HLA-independent triggering of TCR, we expected negligible participation of CD4 in tSV biogenesis. Hence, we selected a guide RNA (gRNA) producing down-regulation of the coreceptor (67.83 ± 10% of baseline) as a control. We also selected gRNAs targeting membrane structural proteins (BST2, CD81) and the ESCRT-I TSG101, which mediates synaptic ectosome budding[5]. We also targeted a disintegrin and metalloproteinase 10 (ADAM10), mediating the

shedding and release of FasL and CD40L trimers from the plasma membrane of activated T cells[43–45]. Knockout of the relevant targets were observed with some residual surface on cells edited for BST2 (45.08 ± 23.19% of control), CD81 (7.63 ± 4.8% of control), and ADAM10 (9.37 ± 2.83% of control; Supplementary Fig. 6a–g). Next, we compared tSV transfer in terms of the Tmax % observed in controls (CD4-gRNA; see Eq. (2) in Methods). While ADAM10 downregulation led to a significant increase in the upregulation of CD40L on the cell surface (224.2 ± 114.8%, Supplementary Fig. 6a, c, h, i) and its synaptic transfer to BSLBs (Fig. 5a), the downregulation of CD81, BST2 and TSG101 (Supplementary Fig. 6f, g, j, k) reduced the transfer of CD40L⁺ tSV without altering the baseline expression levels of CD40L at the plasma membrane (Fig. 5a and Supplementary Fig. 6a, c). TSG101 editing led to a significant reduction on the vesicular transfer of CD40L and TCR without affecting their baseline surface expression levels, indicating a bonafide inhibition of their vesicular sorting dependent on ESCRT-I (Fig. 5a–b). TSG101 downregulation partly phenocopied the effects of ubiquitin depletion by MG132, which affected most substantially the transfer of CD40L⁺ tSV, and to a lesser extent TCR⁺ tSV (Supplementary Fig. 3f–g). CD63⁺ tSV transfer was less affected by downregulation of CD81, ADAM10, BST2, and TSG101 (Fig. 5c), providing an additional piece of evidence for CD63 trans-synaptic transfer relating mostly to the exocytosis of

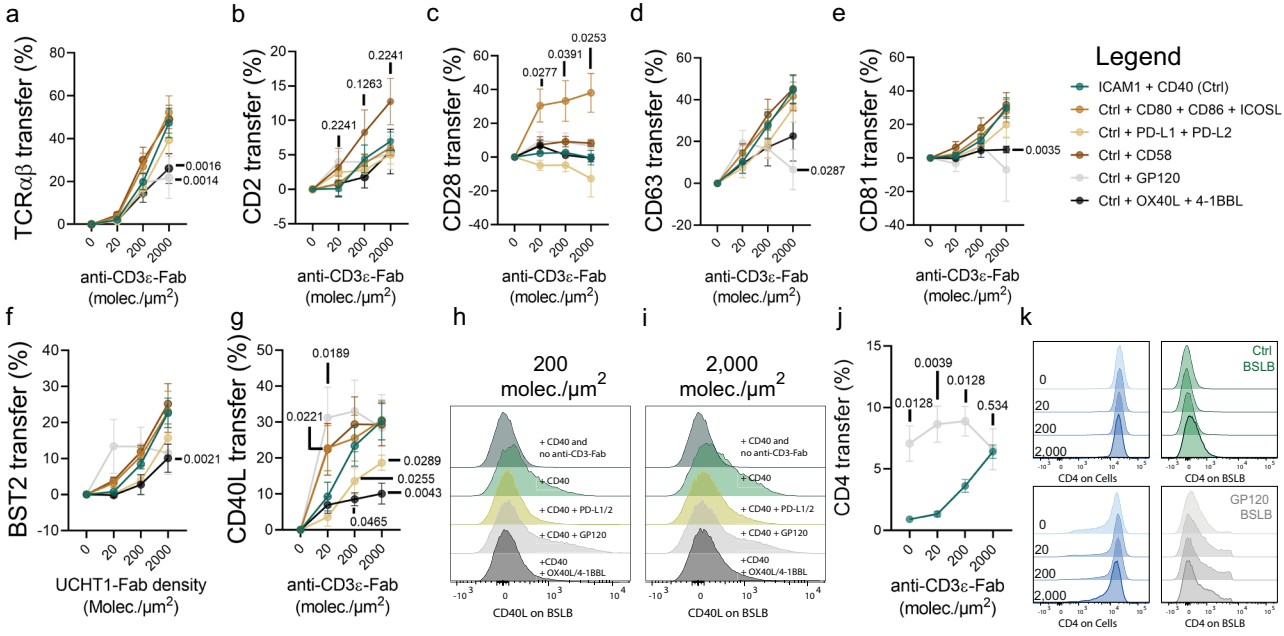

**Fig. 4 Juxtacrine signals influence the tSV output of TH synapses. a–k** Normalized Synaptic Transfer (NST%) of various vesicular proteins in response to BSLBs presenting different juxtacrine signals. Please refer to Supplementary Fig. 1a for a representative gating strategy showing the identification of single BSLBs and T cells by flow cytometry. **a** TCR, **b** CD2, **c** CD28, **d** CD63, **e** CD81, **f** BST2, and **g** CD40L vesicular transfer to control BSLBs (dark green circles) and BSLBs presenting 100 molec./µm² of each juxtacrine signals CD80/CD86/ICOSL (brown circles), CD58 (dark brown circles), PD-L1/PD-L2 (light brown circles), HIV-1 gp120 (light gray circles), and OX40L/4-1BBL (black circles). **h–i** Representative half-overlaid histograms showing CD40L transfer to BSLB with various juxtacrine signals and in the presence of either 200 (**h**) or 2000 (**i**) molec./µm² of α-CD3ε-Fab. **j** NST% of CD4 to control and HIV-1 gp120-presenting BSLBs. **k** Representative histograms showing the loss of CD4 on TH (bottom left panels and blue histograms). Data represents means ± SEM of *n* = 5 (**a–g**, **j**) biologically independent samples across three independent experiments. Normality was determined using Shapiro–Wilk test and statistical significance was determined by two-tailed Multiple *t* test with Holm–Šídák correction ($\alpha = 0.05$) for the multiple comparisons of tSV transfer between control BSLBs and BSLBs decorated with different juxtacrine signals (**a–g**, **j**). Adjusted *P* values are shown next to each significant comparison.

preformed vesicles stored in MVBs. As expected, CD81 and BST2-edited cells showed a significant reduction in the transfer of their respective CD81⁺ and BST2⁺ tSV (Fig. 5d–e, respectively). Imaging of the immunological synapses formed by CRISPR/Cas9-edited cells further revealed that changes in tSV transfer to BSLBs were consistent with the interfacial clustering behavior of CD40L in the synapses of *ADAM10*-, *CD81*- and *TSG101*-edited cells (Fig. 5f–h). *CD81*-, *BST2*-, and *TSG101*-edited cells also showed a reduced Tmax for CD40L (Supplementary Fig. 6l–q). *TSG101*-edited cells also showed reduced Tmax for TCRab (Supplementary Fig. 6l–q) and reduced upregulation of cell surface CD40L upon TCR-triggering (Supplementary Fig. 6q), consistent with the reduced total clustering of CD40L at the cell-SLB interface. Expectedly, *ADAM10*-edited cells showed increased Tmax and reduced α-CD3ε-Fab EC₅₀ for the release of CD40L⁺ tSV (Supplementary Fig. 6l–m), and a marked cell surface upregulation of CD40L following α-CD3ε stimulation (Supplementary Fig. 6q). The partial reduction in CD40L transfer following downregulation of CD81 and BST2, suggests that rather than being essential, these integral membrane proteins work as aiding factors in the release of CD40L⁺, but not TCR⁺ nor CD63⁺ tSV. Also, the differential requirements for TSG101 among TCR⁺, CD40L⁺, and CD63⁺ tSV, which is phenocopied by MG132 treatment, demonstrates different ESCRT-I requirements among tSV. The increased transfer of CD40L⁺ tSV from ADAM10-deficient cells also suggests minimal inclusion of cleaved ectodomains in the transferred pool of CD40L, which is also supported by our immunoblot analyses (Supplementary Fig. 1h).

**tSV carry greater RNA-binding protein content and distinct miRs than EVs.** The assembly of immunological synapses lasting minutes to hours not only facilitates the exchange of juxtacrine signals recruited at the cell-cell interface. Previous studies have suggested a scarce to null load of miR copies in steadily released EVs[46,47], suggesting that the transfer of single miR species via EVs might be less efficient than initially thought and that other modes of communication might favor their intercellular transfer. As observed in previous studies using human T and B-cell lines, stable cell-cell contacts also enable the exchange of miR across the synaptic cleft[3], suggesting that T-cell immune synapses act as facilitators of miR transfer via tSV. To evaluate this hypothesis, we performed first label-free quantifications with liquid chromatography-tandem mass spectrometry (LC-MS/MS) to identify potential RNA-binding proteins (RBPs) enabling miR transfer at the IS. We analyzed tSV and EVs isolated by differential centrifugation of BSLB eluates and cell-cleared culture supernatants, respectively. Since contaminants derived from the elution reagents and procedure might impact the quality of our comparisons, we first identified proteins enriched in eluates derived from activating (α-CD3 Fab⁺) BSLB comparing to null BSLB (coated only with ICAM1 and CD40; Supplementary Fig. 7a). With the aid of Pegasus, we then generated a list of differentially expressed proteins to compare with the EVs from the same TH and CTL cells, respectively (Supplementary Fig. 7b–c). Using average-based normalization we then identified unique and shared proteins within the enriched datasets of tSV and EVs TH (Fig. 6a) and CTL (Fig. 6b, see also Source Data). More detailed gene function analyses of these enriched entities

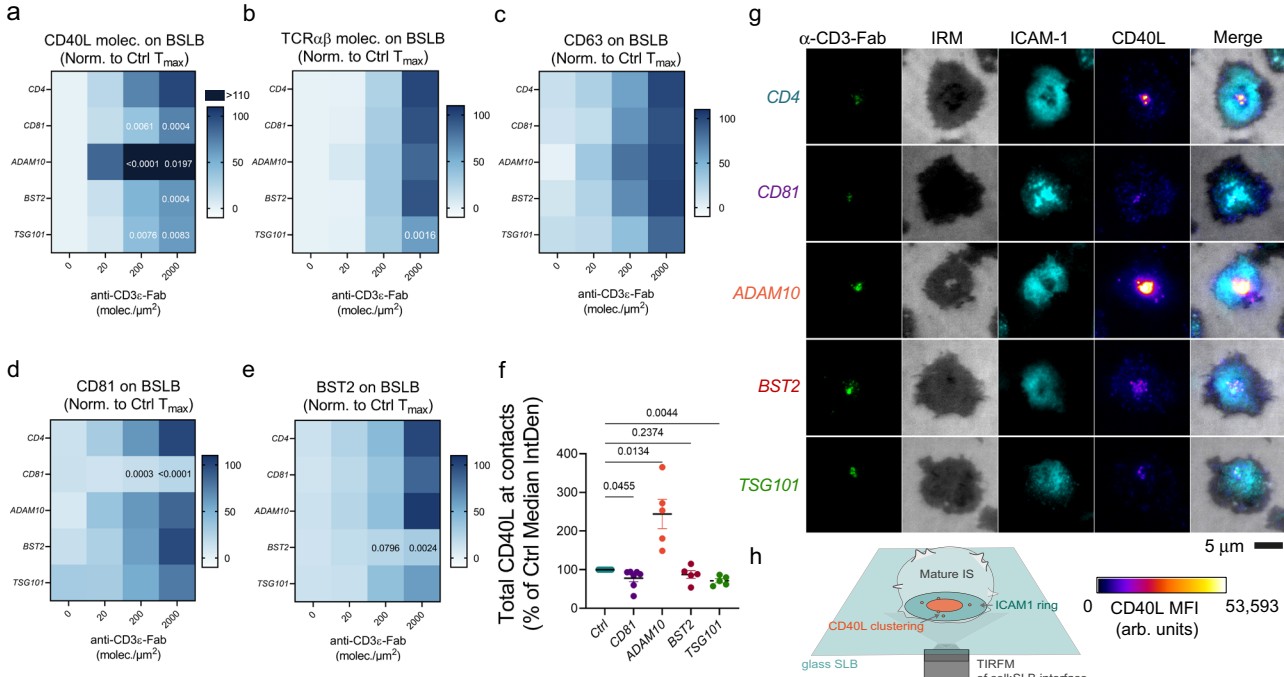

**Fig. 5 ADAM10, TSG101, CD81, and BST2 are key proteins participating in the biogenesis of CD40L⁺ tSV. a–e** Flow cytometry analyses showing the transfer of tSV markers to BSLBs expressed as percent of the maximum transfer (Tmax%) observed in BSLBs co-cultured with *CD4*-edited controls. BSLBs were reconstituted with ICAM1 (200 molec./μm²), CD40 (20 molec./μm²) and anti-CD3ε Fab (0–2000 molec./μm²). Please refer to Supplementary Fig. 1a for a representative gating strategy showing the identification of single BSLBs and T cells by flow cytometry. **a** Heatmaps showing Tmax% for CD40L, **b** TCRαβ, **c** CD63, **d** CD81, and **e** BST2 in BSLB incubated with different CRISPR/Cas9-edited cells (gRNA gene targets shown in italic). **f** TIRFM quantification of normalized medians for the integrated fluorescence intensities of CD40L within synapses formed by CRISPR/Cas9-edited cells. Circles represent the integrated fluorescence median per donor and per CRISPR/cas9 target normalized to its internal control (teal). *CD81*- (violet circles), *ADAM10*- (orange circles), *BST2*- (dark red circles), and *TSG101*-edited cells (green circles) were imaged with TIRFM. **g** Representative TIRFM images (**g**) showing the levels of centrally clustered CD40L in synapses of CRISPR/Cas9-edited cells imaged as depicted in **h** and compared with *CD4*-edited controls. Scale bar = 5 μm. MFI = Mean Fluorescence Intensity, arb. units = arbitrary fluorescence units. IRM = internal reflection microscopy. Data are shown as means ± SEM from n = 10 (**a–e**), and 7 (**f**) biologically independent samples (donors) across five independent experiments. In **a–e**, normality was determined using Shapiro–Wilk test, and statistical significance was determined by parametric two-tailed Multiple *t* test with Holm-Šídák correction (α = 0.05) for the multiple comparisons of Tmax% across different anti-CD3ε Fab densities and between CRISPR/Cas9-edited cells and *CD4*-edited controls (**a–e**), and by one-way ANOVA with mixed-effect analysis and Fisher's LSD test (95% CI) to compare separately each CRISPR/Cas9-edited cells to controls (**f**). Values represent calculated *P* values for each comparison to control.

using PANTHER[48] further revealed that compared with EVs, tSV contained a greater abundance of translational and RNA metabolism proteins in both TH and CTL (Fig. 6c–f). Furthermore, gene set enrichment analyses (GSEA) with PANTHER revealed an enrichment of nucleic acid-binding, including messenger RNA-binding and DNA/RNA helicase activities, in the molecular function categories of tSV compared with EVs (Fig. 6g–j). Consistent with the participation of ESCRT-I on the release of tSV, ubiquitin ligase and ubiquitin processing activities stood out as enriched gene ontology categories in tSV (Fig. 6g–j). Other differences included the negligible detection of ADAM10 and the contaminants Histone 3, LGALS3BP[49], and RNAse 4 in the tSV of TH and CTLs, and a greater detection of known miR targets in tSV (Supplementary Fig. 7d, e, respectively). Importantly, we identified YBX1, a known protein packing small noncoding RNAs (sncRNA) in EVs[50,51] as an enriched protein in tSVs (TH > CTL; Supplementary Fig. 7f), as well as in synaptic ectosomes tethered to sorted BSLBs (with > 3-fold enrichment; see ref. [5]). Other RNA-binding proteins were also commonly identified in independent LC-MS/MS runs of eluted tSV and subpopulations of SE anchored to sorted BSLB (Supplementary Fig. 8a). To validate our LC-MS/MS findings, we then performed TIRFM imaging of TH cell synapses and localized YBX1 and, in a lesser extent, SF3B3[5] in both the synaptic cleft and the proximal membrane of T cells

interacting with α-CD3ε presenting SLB (Supplementary Fig. 8b, c, respectively, compare with isotype control shown in Supplementary Fig. 8d). Also, YBX1 localized in vesicular tracks left behind by cells spontaneously breaking synapse symmetry and resuming migration (Supplementary Fig. 8c, white arrows).

We next sought to evaluate whether the identification of RNA-binding proteins in tSV related to the release of tSV-containing RNA. We used RNASelect[52], a probe fluorogenic upon binding to RNA, to selectively track the mobilization of RNA⁺ structures to the synaptic pole and its release as part of tSV. We also used BODIPY-TR Ceramide to follow membrane structures associated with RNA. Supporting the active release of RBPs, we further found the polarization of ER-like structures and release of ceramide⁺ and RNA⁺ puncta in both the synaptic cleft and in vesicular tracks of live, non-permeabilized TH IS (Supplementary Fig. 8e–f, yellow arrows). As evidenced by the super-resolution imaging with eTIRF-SIM, most RNA puncta lined the periphery of the synaptic cleft as early as 5 min during synapse assembly, indicating the rapid mobilization of RNA⁺ compartments to the contacting membrane of stimulated T cells (Supplementary Fig. 8f, yellow arrows).

The identification of RBP and RNA transfer across the synaptic cleft supports the notion of the IS as a hub facilitating the transfer of RNA effectors, such as miR. Therefore, we sought to evaluate

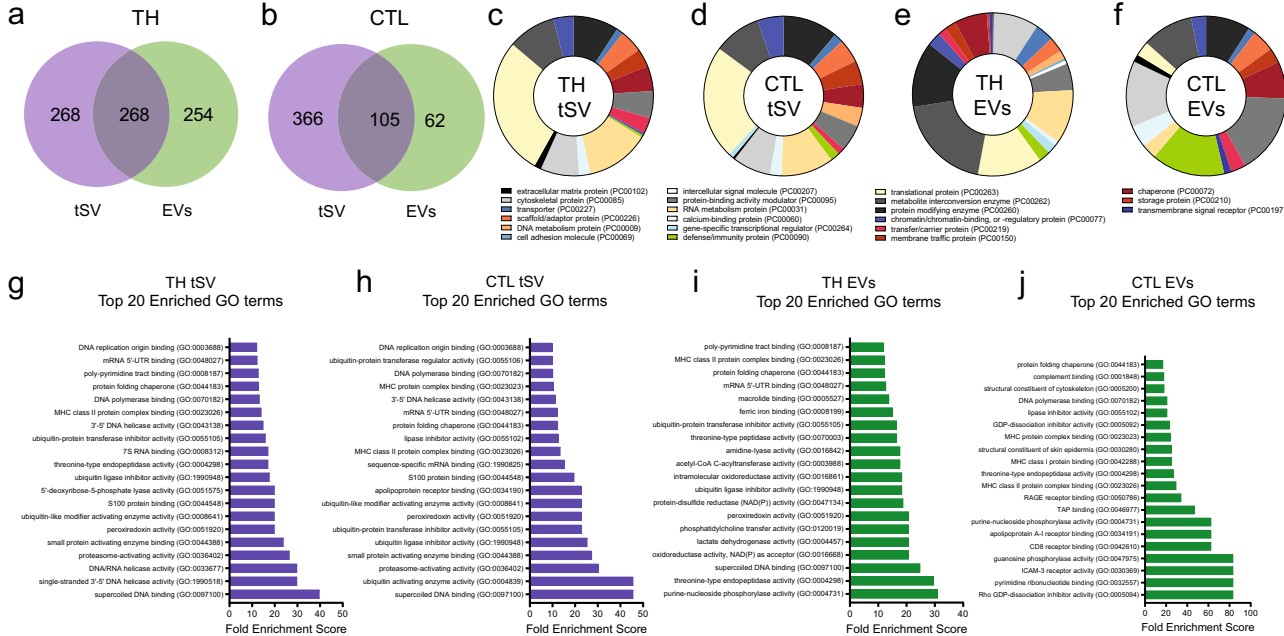

**Fig. 6 tSV are enriched in proteins related to ubiquitination and RNA-binding proteins. a–b** Venn-diagrams showing the overlap of enriched proteins identified in the tSV and EVs of TH (**a**) and CTL (**b**). **c–f** Pie charts showing the enriched protein classes identified with PANTHER[48] in the tSV of TH (**c**) and CTL (**d**), and in their syngeneic EV counterparts (**e–f**). **g–j** GSEA using PANTHER[86] revealed a greater enrichment of molecular function categories related to nucleic acid-binding and ubiquitination in the proteomics of tSV derived from TH (**g**) and CTL (**h**) as compared with syngeneic EVs of TH (**i**) and CTLs (**j**). Data from n = 4 biologically independent samples (donors, **a–j**) across four independent experiments. Statistical analysis for functional and biological pathway enrichment analyses (**c–f**) using PANTHER was performed by the Fisher's exact test and the Benjamini–Hochberg False Discovery Rate (FDR) correction using default Q values. In **g–j**, the result of GO enrichment analyses were filtered using Funset with the Benjamini-Hochberg correction (FDR threshold to 0.05) on P values determined by the hypergeometric test.

whether tSV represents an alternative mechanism of miR transfer by profiling the RNA content deposited on BSLBs. To minimize cell contamination, we isolated BSLB coated with TCR⁺ tSV using fluorescence-activated cell sorting of single BSLBs and single cells after 90 min of co-culture (Supplementary Fig. 9a, b), followed by small RNA purification. Small RNAs were also purified from EVs from the same donors as controls. Prior to comparing the RNA species identified in EVs and tSV, we used the less abundant RNA species found in null BSLBs as background for the identification of miR enriched in tSV (Supplementary Fig. 9c). Analyses of tSV and EVs libraries' raw reads revealed a similar composition of small RNA species, of which miRs were overall more enriched in tSV than EVs and the parental cells (Fig. 7a). After identifying small RNA species conserved across samples, donors, and sequencing runs, GSEA further revealed public and tSV- or EVs-specific miRs in both TH and CTL. Of 200 miR species enriched in TH, 11 were uniquely enriched in tSV and 23 in EVs (Fig. 7b), whereas of 163 miR species enriched in CTL, 17 were unique to tSV and 19 to EVs (Fig. 7c and Supplementary Data 3). Although EVs showed slightly more miR species (Fig. 7b, c), the overall percent enrichment of these small RNA species was higher in tSV (Fig. 7a). Consistent with the wider variety of enriched miR in EVs, we observed a higher variance compared to tSV (Fig. 7d). Analyses using MEME[53] further identified unique and shared miR species in tSV (Fig. 7e, g) and EVs (Fig. 7f, h), which together redundantly associated with immune cell signaling and anti-tumor immunity regulation in both TH and CTLs (Fig. 7i and heatmaps in Supplementary Fig. 9d–g). Congruently, miR-target enrichment analyses revealed a similar degree of redundancy in the targets of enriched miR (see Supplementary Data 4 for the full list of targets and Supplementary Fig. 9h–i for gene ontology categories). Importantly, from the list of predicted targets, AGO2,

GRB2, GRAP2, RRM2, and YBX1 were co-identified in the proteomic datasets of tSV (Supplementary Fig. 7e and 8a), suggesting the transfer of the second layer of information in the form of ribonucleoprotein (RNP) complexes associated to tSV. Altogether, our data suggest that the chain of molecular events unfolding at stable cell-cell contacts facilitates the acute and multimodal delivery of a critical mass of trans-cellular signals.

## Discussion

The first observation of T cells being successfully activated by bead-supported bilayers presenting antigenic MHC-II dates back to 1986[54]. Others have demonstrated the versatility of BSLB as APCs in other experimental contexts, including measuring the dimensions of interfacial receptor-ligand interactions and the localization of lipid species and CD3ε in the interacting pole of T cells[55,56]. Here BSLBs facilitated novel comparisons of tSV and EVs, providing evidence for highly heterogeneous yet unique populations of vesicles being released in the limited dwell-time of the IS. We corroborated quantitative differences in size, protein loads and association to RBPs and miR between tSV and EVs. Furthermore, we also found that the rate of secretion is greater for tSV compared with EVs. We have previously shown that TH IS shed an approximate of 25–30 TCRαβ⁺ synaptic ectosomes after 20 min of synapse formation on SLB[5,6]. In contrast, the quantification of steadily released TCRαβ⁺ EVs by NanoFCM evidenced their rather low constitutive secretion ranging from a minimum of 0.54 to a maximum of 8.96 EVs/cell over a period of 48 h of culture (please refer to Source Data).

The acute pharmacological inhibition and CRISPR/Cas9-editing of TH strongly indicated that, mechanistically, tSV heterogeneity is explained by contributions of several cellular types of machinery recruited at the interacting pole of activated T cells. We provide evidence for an overarching hierarchy in the

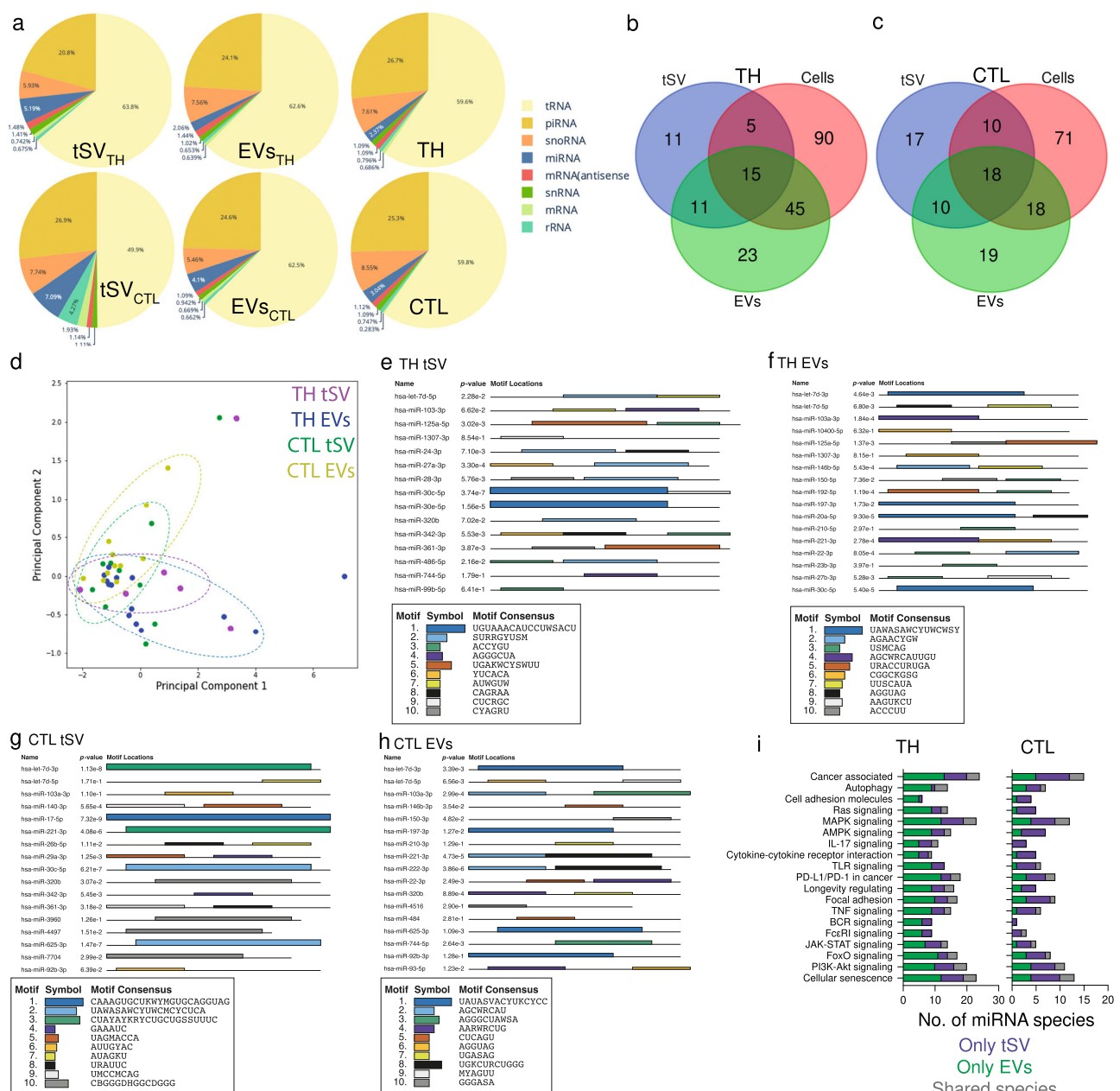

**Fig. 7 tSV are enriched in miRs with considerable functional equivalence to EVs miRs. a** Abundance of different genome-mapped RNA species was found associated with tSV, syngeneic EVs, and parental TH or CTL cells. **b**–**c** Venn diagram showing the overlap of enriched miR species associated with tSV, EVs, and parental cells for TH (**b**) and CTL (**c**) (see also Supplementary Data 3). **d** miR heterogeneity is further reflected by PCA of the analyzed tSV and EVs. **e**–**h** MEME analyses revealed consensus motifs (central inserts) mapped in miRs of TH tSV (**e**), TH EVs (**f**), CTL tSV (**g**), and CTL EVs (**h**). **i** Number of miR species enriched uniquely in tSV (violet), uniquely in EVs (green), or shared (gray) annotated according to their GO categories. MiR targets are summarized in Supplementary Data 4. Data from $n = 8$ biologically independent samples (donors, **a**-**i**) from six independent experiments. For functional and biological pathway enrichment analyses using MIENTURNET (s) $P$ values were determined by hypergeometric statistic test and FDR ($Q = 0.05$).

mechanisms participating in the vesicular transfer of CD40L, with dominant participation of ER/TGN transport and ESCRT machinery for its acute mobilization to the plasma membrane, and secondary involvement of MVB and lysosomes. The selective increase in TCR$^+$ tSV transfer resulting from dynamin inhibition and the reduced effect of ESCRT-inhibition/downregulation on CD63$^+$ tSV transfer provide additional evidence for different sorting mechanisms participating in the biogenesis of tSV (Supplementary Figs. 3f–i and Fig. 5). We provide evidence for a critical role of ADAM10, a protease known to be active at the plasma membrane, in regulating the pool of surface CD40L and

the resulting levels of CD40L$^+$ tSV transferred dependent on CD40. Interestingly, our LC-MS/MS analyses indicate that compared to tSV, EVs incorporate a higher amount of ADAM10 (Supplementary Fig. 7d), explaining their reduced CD40L content (Fig. 1k-l).

Microvilli lead the formation of effector membranous particles referred to as synaptosomes enriched in LFA-1, CD2 and cytokines[57]. Microvilli tips accumulate proteins including TCR, CD2, and CD4[57,58]. Here we show that T cells transfer little vesicular CD2 and CD4 (Figs. 1b and 2b–c), which increased at limited levels only upon ligation by CD58 and HIV-1 gp120,

respectively (Fig. 4b, j–k). Also, TCR signaling, which is required for tSV biogenesis, is known to produce microvilli disassembly[58], suggesting that synaptosomes differ in origin and structure from synaptic ectosomes and the overall output of tSV. Rather, synaptosome-like structures might mediate the negligible release of extracellular vesicle-like material to BSLBs sustaining cell adhesion but devoid of antigen/α-CD3 Fab (Fig. 1k).

TSV biogenesis is a highly dynamic process connecting the integration of signals presented by APCs with the particulate output of activated T cells. Earlier electron microscopy studies demonstrated budding of TCR⁺ SE with partial polarization of MVB and TGN in stimulated cells (Fig. 1c in ref. [6]). In these experimental settings, SLB lacked CD40, which as we later demonstrated, promotes quick mobilization of CD40L to the plasma membrane and into budding SE as microclusters[5]. Here we provide evidence that shedding of CD40L⁺ tSV requires CD40 on presenting BSLBs and the rapid relocation of CD40L from the ER-Golgi, which is readily detected by 90 min of interaction between T cells and BSLBs (Supplementary Figures 4b and 6q). We provide evidence of other signals finely tuning tSV heterogeneity, including B7 receptors (CD80/86), PD-1 ligands, other TNFSF members, and non-native ligands. For instance, HIV-1 gp120 promoted the vesicular transfer of CD4 and CD40L at low physiological densities of α-CD3ε Fab. Similarly, the non-native interaction between CARs and antigenic HLA complexes led to a remarkably efficient transfer of CAR as part of tSV and at the expense of its cell surface expression levels (Supplementary Fig. 5d, red arrow). The superior vesicular shedding of CAR containing CD3ζ signaling tails might result from rewired TCR signaling networks and the formation of multiple short-lived synapses[59]. Since reduced CAR expression might desensitize CART to new encounters with target cells, more research is needed to understand the impact of CAR release on the therapeutic effectiveness of CART therapies.

BSLBs made possible the first side-by-side comparison of the protein and sncRNA composition of tSV and EVs. While shared and vesicle-specific miR species were identified in tSV and EVs, RBPs were significantly more enriched in tSV (Fig. 6g–j and Supplementary Fig. 7e–f). We speculate that the high level of target redundancy found in enriched miRs of tSV and EVs might synergize in the functional modulation of recipient cells. If true, such synergism among miR could circumvent the limitation imposed by only few copies of single miR species transferred associated with EVs[46,47]. Additionally, the acute delivery of tens of vesicles at the immune synapse might help promote the delivery of a critical mass of miRs and their partner proteins to effectively interfere target mRNAs. How the integration of juxtracrine signals, such as antigen density (and TCR affinity) and co-stimulation/co-repression, shapes the composition and loads of miR and RBPs transfer across cell-cell contacts requires further investigation.

Since miR targets participate in cell cycle regulation, senescence, and immune receptor signaling, the release of tSV might also promote the clearance of miR species controlling various aspects of the T-cell effector response. Interestingly, telomeric binding proteins were also found among the enriched GO terms of TH tSV (please refer to Source Data of Fig. 6g–j) and the differential expression analyses of synaptic ectosome proteins[5]. Whether the transfer of miRs, and to a lesser extent telomeres, in tSV contributes to the paracrine regulation of senescence, as preliminarily shown for APCs transferring telomeres[60], remains to be elucidated.

The intrinsic feed-forward function of tSV-enriched immune receptors and miRs, specifically and timely released in response to the antigen is different in kind and structure to other signals integrated during early T-cell activation. Therefore, we consider that the heterogeneity and effector diversity of the particulate output of T-cell synapses, including tSV and SMAPs, deserves its functional classification as signal four. We expect BSLB will help others in the study of other key drivers of signal four-dependent communications. Here we provide numerous examples of how BSLBs can be tailored to model the effects of different juxtacrine signals on the composition and release of tSV. However, compared to real APCs, BSLBs lack the trogocytic and trans-endocytic machinery required to engulf large and small membrane fragments across the synaptic cleft[7–11,61]. While this has the advantage of specifically addressing molecules transferred as part of T-cell tSV, is limited in predicting the exchange of molecules resulting from phagocytic cells nibbling the plasma membrane of T cells during transient contacts.

Modeling different immune synapses and tSVs also requires considering the kinetics of interaction between T cells and different APCs, such as B cells and DCs. In vivo imaging studies have shown that compared to the interaction with DCs, which last up to several hours[62–64], the interaction between T cells and B cells last from a few minutes to an hour[65]. BSLBs can be co-cultured with T cells in different time scales from 1 (Fig. 3c) to 24 h (Supplementary Fig. 10a–c). In fact, in BSLB-T-cell co-cultures mimicking prolonged DC-T-cell interactions, BSLB efficiently capture T-cell membrane alongside the tSV markers TCR and CD81 (Supplementary Fig. 10a–c). In comparison, monocyte-derived DCs (moDCs) co-cultured with T cells acquire T-cell trans-synaptic material (i.e., antigen-driven) within minutes of synapse formation (Supplementary Fig. 11a–b), but also membrane fragments consistent with the engulfment of EVs and trogocytosis. After 24 h of culture, antigen-null moDCs engulfed TH membrane fragments despite failing at inducing the formation of stable and polarized contacts with TH cells (Supplementary Fig. 11c–e). In contrast, after 24 h of co-culture, antigen-dependent contacts led to a greater acquisition of T-cell membrane and significant activation of moDCs as evidenced by the upregulation of ICAM1, CD86, and CD40 (Supplementary Fig. 11f–i). Remarkably, the latter replicated the activation of moDCs using synthetic tSV carrying physiological densities of CD40L[5] and correlated with significantly higher CD40L surface expression on moDCs (Supplementary Fig. 11i), suggesting that activation resulting from the acquisition of CD40L⁺ tSV is superior to that resulting from trogocytosis or the uptake of constitutive EVs in cell-cell culture systems. Thus, our findings suggest that tSV are superior cell-cell communication entities, highlighting the relevance of their direct analysis and modeling with BSLBs.

Undoubtedly, the comparison of EVs and tSVs in other immune cells known to secrete highly heterogeneous EVs, such as DCs[66], or in cells using similar interfacial communications is needed to elucidate whether tSV are pivotal information units of cellular networks. Pathogen evolution has co-opted the cellular machineries mediating tSV release. For instance, the HIV-1 Gag protein facilitates the transfer of viral information among interacting cells by co-opting the budding machinery of synaptic ectosomes[6,67,68]. We expect BSLBs will also help unraveling new pathogenic determinants hijacking the cellular machinery required for tSV shedding, broadening our understanding of 'trojan horses'[67] driving infectious diseases.

## Methods

**Ethics**. Different T-cell populations were isolated from leukapheresis reduction system (LRS) chambers from de-identified, non-clinical healthy donors. The non-clinical issue division of the National Health Service and the Inter-Divisional Research Ethics Committee of the University of Oxford (RECs 11/H0711/7 and

R51997/RE001, and Oxford Biobank Application number 18/A019; IRAS Project ID:66954) approved the use of LRS chambers containing blood from anonymized donors (with unknown age and sex) and under written consent.

**Isolation and expansion of human CD4+, CD8+, and Treg cells from peripheral blood**. Briefly, human T cells were isolated from LRS-concentrated peripheral blood by negative immunodensity selection using either CD4+, CD8+ or CD4+CD127low (RosetteSep, StemCell Technologies, #15022, #15023, and #15361). Enriched CD4+ CD127low cells were immediately used for fluorescence-activated cell sorting of regulatory T cells (Treg) based on low CD127 fluorescence and high CD25 fluorescence. Briefly, Tregs were sorted from enriched CD4+ CD127neg peripheral blood cells using a FACSAria III cell sorter. A nested gating strategy was used in which CD4+ CD127neg CD25high cells were gated such that only the brightest 5% of CD25high cells were sorted (Supplementary Figure 5a). Recovered cells were stimulated with Human T-activator CD3/CD28 Dynabeads (Thermo-Fisher) at a bead-to-cell ratio of 3:1 and 1000 U/mL of recombinant human IL-2 (Peprotech). Fresh IL-2 containing media was added every 48 h and Human T-activator CD3/CD28 Dynabeads were added again at day 7 and removed at day 15 of expansion. This protocol allowed us the enhanced recovery of FoxP3+ T cells by day 15 of expansion (see Supplementary Fig. 5b–c). On the other hand, conventional CD4+ and CD8+ T cells were expanded using a bead-to-cell ratio of 1:1 and 100 U/mL of IL-2. After 3 days of activation, Human T-activator CD3/CD28 Dynabeads were removed and the conventional CD4+ and CD8+ cells kept in complete RPMI media supplemented with 100 U/mL of IL-2. All BSLB synaptic transfer experiments were performed after preconditioning of T cells to IL-2-depleted complete RPMI 1640 media containing 10% of either heat-inactivated AB human serum (Treg & controls) or fetal bovine serum, and 100 μM non-essential amino acids, 10 mM HEPES, 2 mM L-glutamine, 1 mM sodium pyruvate, 100 U/ml of penicillin and 100 μg/mL of streptomycin).

**Culture of CD4+ T-cell clones**. The HLA-DRB1*09:01-restricted T-cell clones 16, 35, and 40 (all specific against the influenza H3 HA$_{338–355}$ peptide: NVPEKQTRGIFGAIAGFI) were expanded using a ratio of 1 clone: 2 feeder cells (irradiated, pooled PBMCs from 2–3 healthy donors) at a total cell concentration of $3 \times 10^6$ cells/mL in RPMI 1640 supplemented with 10% heat-inactivated AB human serum and 30 μg/ml of PHA for three days. Then, 100 U/ml of recombinant human IL-2 were added to fresh media, which was replaced every 2 days. Clones were used between days 8 and 12 of culture.

**Production of chimeric antigen-receptor (CAR) recombinant CD8+ T cells**. The CAR constructs that bind the NY-ESO-1$_{157–165}$ HLA-A2 complex were a kind gift from Cristoph Renner (Zurich, Switzerland)[69,70]. Briefly, the T1 CAR binds the C9V NY-ESO-1 APL with a $K_D$ of ~2 nM. The CAR construct was packaged within third-generation self-inactivating lentiviral transfer vectors with the EF1α promoter and the Woodchuck Hepatitis Virus Post-Transcriptional Regulatory Element. The CAR features an scFv binding domain, Ig domain spacers derived from human IgG1, the CD28 transmembrane domain, and the CD3ζ signaling tail. For lentiviral production, 293 T cells (ATCC CRL-3216) were co-transfected with a mix of the VSV-G (370 ng), the lentiviral CAR (800 ng), and RSV-Rev and GAG (950 ng each) plasmids. The plasmid mix was prepared in DMEM (Merck, #D6429) containing X-treme Gene HP DNA Transfection Reagent (Merck, #6366546001) and after 20 min preincubation at RT, the mix was added to 293 T cells in a dropwise manner. Twenty-four h after isolation and stimulation with Human T-activator CD3/CD28 Dynabeads, CD8+ T cells were transduced with freshly harvested and 0.45 μm-filtered supernatant containing 50 U/ml of recombinant human IL-2. Three days after transduction Dynabeads were removed and the cells were kept in the culture at a concentration of $1 \times 10^6$ cells/mL in IL-2 supplemented RPMI media (see above). Transduction efficiency was determined between days 8–10 of culture using a polyclonal goat anti-human IgG Fc PE (ThermoFisher Scientific, #12-4998-82) and the synaptic transfer to BSLB was addressed immediately.

**Bead-supported lipid bilayers (BSLB)**. Non-functionalized silica beads (5.00 ± 0.05 μm diameter, Bangs Laboratories, Inc.) were washed extensively with PBS in 1.5 ml conical microcentrifuge tubes. BSLBs were formed by incubation with mixtures of liposomes to generate a final lipid composition of 0.2 mol% Atto-DOPE; 12.5 mol% DOGS-NTA in DOPC at a total lipid concentration of 0.4 mM. In this work, we used DOPE lipids conjugated with either Atto 390, 488, 565, or 647. The resultant BSLB was washed with 1% human serum albumin (HSA)-supplemented HEPES-buffered saline (HBS), referred to herein as HBS/HSA buffer. To saturate NTA sites, BLSB were then blocked with 5% casein 100 μM NiSO$_4$ for 20 min. After two washes, BSLB were loaded with concentrations of His-tagged proteins required to achieve the indicated molecular densities (in the range of 1–100 nM; please refer to each figure legend). Excess proteins were removed by washing with HBS/HSA after 30 min. T cells ($2.5 \times 10^5$/well) were incubated with BSLB at 1:1 ratio in either U-bottomed or V-bottomed 96 well plate (Corning, #3367 or #3894, respectively) for 90 min at 37 °C in 100 μl HBS/HSA. For gentle dissociation of BSLB: cell conjugates, culture plates were gradually cooled down by incubation at RT for 15 min, followed by incubation on ice. After 45 min, cells and BSLB were centrifuged at $300 \times g$ for 5 min prior to resuspension in ice-cold 5%

BSA in PBS pH 7.4. Single BSLB and cells were gently resuspended prior to staining for flow cytometry analysis or sorting.

**Multicolor flow cytometry (FCM)**. Staining with fluorescent dye conjugated antibodies was performed immediately after the dissociation of cells and BSLB conjugates. Staining was performed in ice-cold 5% BSA in PBS pH 7.4 (0.22 μm-filtered) for a minimum of 30 min at 4 °C. Then, cells and BSLB were washed three times and acquired immediately using an LSRFortessa X-20 flow cytometer equipped with a High Throughput Sampler (HTS). For absolute quantification, we used Quantum Molecules of Equivalent Soluble Fluorescent dye (MESF) beads (see below), which were first acquired to set photomultiplier voltages to position all the calibration peaks within an optimal arbitrary fluorescence units' dynamic range (between $10^1$ and $2 \times 10^5$, and before compensation). Fluorescence spectral overlap compensation was then performed using single color-labeled cells and BSLB, and unlabeled BSLB and cells. For markers displaying low surface expression levels unstained and single color stained UltraComp eBeads (ThermoFisher Scientific Inc.; #01-2222-42) were used for the calculation of compensation matrixes. After application of the resulting compensation matrix, experimental specimens and Quantum MESF beads were acquired using the same instrument settings. In most experiments acquisition was set up such as a minimum of $5 \times 10^4$ single BSLBs were recorded. To reduce the time of acquisition of high throughput experiments a minimum of $1 \times 10^4$ single BSLBs were acquired per condition instead. All flow cytometry and immunoblot antibodies described in this study are listed in Supplementary Table 1 and in the Nature Publishing Group Reporting Summary and includes information on catalog numbers and effective concentrations or dilutions used. Analysis of flow cytometry data was performed with FlowJo LLC (v10.8.1).

The percent normalized synaptic transfer was calculated using Eq. (1):

$$NST\% = \frac{BSLB\ cGMFI}{BSLB\ cGMFI + Cells\ cGMFI} \times 100 \qquad (1)$$

The percent transfer maximum was calculated using Eq. (2):

$$Tmax\% = \frac{BSLBs\ cGMFI}{BSLB\ cGMFI\ max} \times 100 \qquad (2)$$

For both calculations, the corrected GMFI (cGMFI) for cells and BSLBs was calculated by subtracting the signal derived from either isotype control-labelled BSLBs or antigen-null BSLBs. In some experiments, the cGMFI was replaced with absolute molecules and the transfer expressed as Tmax% of molec./BSLB.

**Calibration of flow cytometry data**. T cells and BSLB were analyzed using antibodies with known fluorescent dye to Ab ratio (F/P) in parallel with the Quantum MESF beads (Bangs Laboratories, Inc. IN, USA), allowing the calculation of the absolute number of antibodies bound per T-cell and per BSLB after subtraction of unspecific signals given by isotype control antibodies. We used MESF standard beads labeled with the Alexa Fluor® dyes 488 and 647 to estimate number of dye molecules from mean fluorescence intensities (corrected and geometric, cGMFI). Briefly, MESF beads provided five different populations of beads with increasing GMFI allowing the linear regression of corrected GMFI (cGMFI) over MESF. The resulting slope is then used for the interpolation of total fluorochromes bound to either BSLB or cells from cGFMI values. Since we also used antibodies with known fluorochromes per protein (F/P), we then estimated the number of bound antibodies (and hence molecules) per BSLB by dividing the estimated fluorochrome number by the detection antibody F/P value (Number of molec./event = Fluorescent molec.$_{event}$/(F/P)$_{Ab}$).

**Trans-synaptic vesicle elution from BSLBs**. Cells and BSLBs were incubated at a 1:1 ratio for 90 min at 37 °C and 5% CO$_2$ in Phenol Red-free FBS-free RPMI 1640 supplemented with 100 μM non-essential amino acids, 2 mM L-glutamine, 1 mM sodium pyruvate, 100 U/ml of penicillin and 100 μg/mL of streptomycin. Cultures needed to be scaled up and therefore a CO$_2$ incubator and supplemented FBS-free, Phenol Red-free RPMI 1640 was used instead of 1%HSA/HBS. Cells and BSLBs were collected at different time points of the elution protocol to control with FCM both the transfer of material from T cells to BSLB, and the release of tSV from BSLB upon addition of EDTA for the chelation of Ni. Briefly, elution was performed as follows; first conjugates were cooled down 15 min at RT and then 40 min on ice to gentle separate cells and BSLB. Then, conjugate suspensions were resuspended by adding three volumes of ice-cold PBS pH 7.4 and centrifuged at $100 \times g$ for 1 min and +4 °C to enrich for BSLB. After two washes with ice-cold PBS pH 7.4, BSLB were resuspended in ice-cold 50 mM EDTA in PBS pH 7.4 for 2 h. BSLB-free supernatants were then centrifuged at +4 °C in sequential steps at $300 \times g$ for 5 min (twice), then at $2000 \times g$ for 10 min and $10,000 \times g$ for 10 min. Finally, recovered supernatants were centrifuged at $120,000 \times g$ for 4 h at +4 °C and the pellets washed once more with PBS pH 7.4 and kept at +4 °C or frozen at −80 °C until analyses by either NTA, NanoFCM or immunoblotting. Steadily released EVs were isolated by the same differential centrifugation procedure indicated above. To minimize serum and debris contamination, T cells were cultured in OptiMEM-I supplemented with 100 U/mL of IL-2, 100 μM non-essential amino acids, 2 mM L-glutamine, 1 mM sodium pyruvate, 100 U/ml of penicillin, and 100 μg/mL of streptomycin for no longer than 48 h. All samples isolated by UC

were resuspended in 0.22 μm-filtered PBS pH 7.4 to reduce background signals in downstream particle measurement analyses with either nanoFCM or NTA.

**TEM**. Negative staining of thinly spread vesicle preparations for transmission electron microscopy (TEM) was prepared as follows;[71] briefly, carbon support film-coated 3 mm copper grids (300 mesh) were plasma treated for 20 s using a Leica EM ACE200 Vacuum Coater. Then, 10 μL of isolated vesicle populations were deposited on and incubated at RT for 5 min, followed by removal of the excess sample with a Whatman N°1 paper and staining with 2% uranyl acetate for 10 s at RT. After removal of excess uranyl acetate, the samples were dried for 10 min and analyzed using a Tecnai 12 TEM at 120 kV using a Gatan OneView CMOS camera. A final magnification of ×29,000 was used for imaging isolated vesicle populations.

**NTA**. Eluted tSV and purified EV preparations were resuspended in 0.22 μm-filtered PBS pH 7.4 in a 1:100 dilution and kept on ice for NTA. The instrument used for NTA was Nanosight NS300 (Malvern Instruments Ltd) set on light scattering mode and instrument sensitivity of 15. Measurements were taken with the aid of a syringe pump to improve reproducibility. Three sequential recordings of 60 s each were obtained per sample and NTA 3.2 software was used to process and average the three recordings to determine the mean size.

**Nano-flow cytometry**. Flow NanoAnalyzer model type N30E (NanoFCM Inc., Xiamen, China) that allows single exosomes detection was used to determine the size distribution and granular concentration of EVs. The Nano-flow cytometry analysis was performed using the Flow NanoAnalyzer (NanoFCM Co., LTD) according to manufacturer's instructions. The Silica Nanospheres Cocktail (S16M-Exo, NanoFCM) was employed as the size standard to construct a calibration curve to allow the conversion of side scatter intensities to particle size. A concentration standard (200 nm PS QC beads, NanoFCM) was used to measure particle concentration. The laser used was a 488 nm laser at 25/40 mW, 10% ss decay. Lens filters equipped were 525/40 (AF488) and 580/40 (PE). Before staining samples were acquired to determine particle concentration such that a total of $10^8$ vesicles of either tSV or EVs were labeled per condition. Before use, fluorochrome-conjugated antibodies were spun down at $10,000 \times g$ for 10 min at $+4\,°C$. Isotype control antibodies were used at the same effective concentrations (ranging from 0.2 to 5 μg/mL) and incubation times. Staining antibodies were conjugated to AF488 and AF647. Staining was performed for 30 min on ice. Samples were washed and centrifuged for 1 h at $100,000 \times g$ and $+4\,°C$. Labeled vesicles were then resuspended in 50 μL of PBS. Buffer alone (PBS), unstained vesicle, isotype controls, and auto thresholding were used to define positivity.

**Planar supported lipid bilayers (PSLB)**. Liposome mixtures were injected into flow chambers formed by sealing acid piranha and plasma-cleaned glass coverslips to adhesive-backed plastic manifolds with 6 flow channels (StickySlide VI 0.4; Ibidi). After 30 min the channels were flushed with HBS-HSA without introducing air bubbles to remove excess Liposomes. After blocking for 20 min with 5% casein supplemented with 100 μM $NiCl_2$, to saturate NTA sites, followed by washing and then His-tagged proteins were incubated on bilayers for an additional 30 min. Protein concentrations required to achieve desired densities on bilayers were calculated from calibration curves constructed from FCM measurements on BSLB and analyzed alongside MESF beads (MESF; Quantum™ Bangs Laboratories Inc.). Bilayers were continuous liquid disordered phase as determined by fluorescence recovery after photobleaching with a 10 μm bleach spot on an FV1200 confocal microscope (Olympus).

**Immunological synapse formation on glass-SLB**. Prior to immunological synapse imaging, primary T-cell blasts were washed twice and resuspended in prewarmed HBS/HSA buffer to a final concentration of $5 \times 10^6$ cells/mL. Then, $5 \times 10^5$ T cells were stimulated for 20 min at 37 °C and 5% $CO_2$ over PSLB containing 30 molec./μm² anti-CD3ε Fab (clone UCHT-1; Alexa Fluor® (AF) 488 or unlabeled), 200 molec./μm² of ICAM1 AF405, and 20 molec./μm² of CD40 (AF488 or unlabeled). Then, cells were stained with 1 μg/mL of wheat germ agglutinin (WGA) conjugated to CF568 (Biotium; #29077-1) and 1 μg/mL anti-CD154 (CD40L) clone 24–31 AF647 for 15 min at RT and in the dark. Cells were washed three times with 5% BSA 20 mM HEPES 2 mM $MgCl_2$ in PBS and fixed with prewarmed 4% electron microscopy grade formaldehyde in PBS pH 7.4 containing 2 mM $MgCl_2$ for 10 min at RT and in the dark. After two washes, cells were kept in HBS/HSA until imaging by TIRFM. For intracellular staining, cells were permeabilized for 3 min with 0.1% Triton X-100 in PBS, washed and blocked for 60 min with 5% BSA in PBS before staining with primary antibodies (1 μg/mL for 1 h at RT). Primary antibodies included Rabbit anti-human TSG101 clone EPR7130B AF647 (#ab207664), anti-human YB1 (YBX1) clone EPR22682-2 (#ab255606), anti-human SF3B3 clone EPR18440 (#ab209402), and Rabbit Isotype control EPR25A AF647 (#ab199093). After three washes, cells were blocked with 0.22 μm-filtered 5% Donkey Serum for 1 h at RT before staining for 1 h with AF647 Affi-niPure F(ab')₂ Fragment Donkey Anti-Rabbit IgG (H+L) (Jackson ImmunoR-esearch, #711-606-152). After four washes, cells were washed and then stained with anti-Vimentin clone EPR3776 AF555 (#ab203428). Cells were washed four times before acquisition. To track the mobilization of RNA and membrane containing

structures within the Immune Synapse, T cells were labeled with BODIPY-TR Ceramide complexed to BSA and SYTO RNASelect on days 7 to 10 of culture. Briefly, T cells were washed twice in prewarmed PBS pH 7.4 and resuspended in PBS containing 10 mM HEPES and 5 μM of SYTO RNASelect (Invitrogen, #S32703) and 170 μg/mL of BODIPY-TR Ceramide complexed to BSA (Invitrogen, #B34400). After incubation for 20 min at 37 °C, cells were washed twice with 10 volumes of fully supplemented Phenol Red-free RPMI 1640 (Gibco, #11835-063) and centrifugation steps of $300 \times g$ for 10 min at RT. BODIPY-TR and SYTO RNASelect-labeled T cells were rested a minimum of 4 h and then washed with 10 volumes of HBS/BSA buffer before imaging on PSLBs containing anti-CD3ε Fab (30 molec./μm², unlabeled), ICAM1 (200 molec./μm²) (unlabeled), and CD40 (20 molec./μm²; unlabeled). RNA mobilization at the IS was imaged using total internal reflection fluorescence microscopy (TIRFM) with excitation at 488 nm.

**TIRFM**. TIRFM was performed on an Olympus IX83 inverted microscope equipped with a 4-line (405 nm, 488 nm, 561 nm, and 640 nm laser) illumination system. The system was fitted with an Olympus UApON 150 × 1.45 numerical aperture objective, and a Photometrics Evolve delta EMCCD camera to provide Nyquist sampling. Quantification of fluorescence intensity was performed with Fiji/ImageJ (National Institute of Health) and MATLAB R2019b. A batch measure macro was used to automatically segment cell:SLB contacts based on internal reflection microscopy followed by both background subtraction and the measure of fluorescence across different channels.

**Spectral flow cytometry of T-cell-DC synapses**. Monocytes and TH cells were isolated from peripheral blood packed in LRS chambers from de-identified, healthy donors. T cells were activated and blasted as indicated above. Monocytes were isolated using negative immunodensity Human Monocyte Enrichment Cocktail (RosetteSep, StemCell Technologies, #15068) following manufacturer's guidelines and then cultured for 7 days with 50 ng/mL of recombinant human interleukin-4 (IL-4, Peprotech, #200–04 A) and 100 ng/mL granulocyte-monocyte colony-sti-mulating factor (GM-CSF, Immunotools, #11343125). In some experiments, T cells were labeled on day 6 with 170 μg/mL of BODIPY-TR Ceramide complexed to BSA (Invitrogen, #B34400) in PBS containing 10 mM HEPES for 20 min at 37 °C. Cells were then washed twice with 10 volumes of fully supplemented Phenol Red-free RPMI 1640 (Gibco, #11835-063) and centrifugation steps of $300 \times g$ for 10 min at RT. BODIPY-TR labeled T cells were rested overnight before co-culturing with monocyte-derived DCs (moDCs). On day 7 of culture, half of the moDCs were incubated for 1 h at 37 °C and 5% $CO_2$ with 100 ng/mL of Staphylococcal Super-antigen B (SEB)[72]. Then, moDCs and T cells were counted, washed and co-cultured at a 1:1 ratio for either 90 min or 24 h at 37 °C and 5% $CO_2$. T cells and moDCs were cooled to RT for 15 min and then incubated on ice for another 30 min. Then, co-cultures were centrifuged at $500 \times g$ for 5 min at $+4\,°C$ and then resuspended in 0.22 μm-filtered $Ca^{2+}$ and $Mg^{2+}$-free PBS (Gibco, #10010-015) containing 10% FBS (heat inactivated), 2 mM EDTA and 1:50 dilution of Human TruStain FcX Fc Receptor blocking solution (BioLegend, #422302). After a minimum of 15 min of incubation on ice, cells were stained for a minimum of 30 min on ice with a pre-optimized staining panel containing anti-human CD40 PercPCy5.5 clone 5C3 (BioLegend, #334316), anti-human CD4 BV650 clone OKT4 (BioLegend, #317436), anti-human CD40L AF647 clone 24–31 (BioLegend, #310818), anti-human CD54 (ICAM1) BV711 clone HA58 (BD Horizon, #564078), anti-CD11c APC-Cy7 clone Bu15 (BioLegend, #337218), anti-human CD86 BV785 clone IT2.2 (BioLegend, #305442). Cells were washed three times using ice-cold PBS and acquired immediately using the Aurora Spectral Flow Cytometer and the Spec-troFlo acquisition software (Cytek Biosciences) equipped with 5-lasers (UV/V/B/YG/R) and an automated sample loader. Analysis of flow cytometry data was performed with FlowJo LLC (v10.8.1).

**Airyscan confocal microscopy**. Airyscan imaging of T-cell and SLB was per-formed on a Zeiss Axio Observer.Z1 LSM 980 confocal laser-scanning microscope equipped with an Airyscan 2 module (Zeiss, Oberkochen, Germany) consisting of 32 concentrically arranged GaAsP PMT detectors and 2 MA-PMT channels. The acquisition was performed using the Airyscan super-resolution (SR) and best signal Smart Setup and a C Plan-Apochromat ×63/1.4 NA Oil DIC magnification objective. Illumination was provided by a Solid-State Light Source Colibri 7 LED lamp and by Diode lasers at 639 nm, 594 nm, and 488 nm with 0.4% laser power and 850 V detector gain for all channels. The imaging field was defined using a 6.0× scan zoom (crop area) and a Z-coverage spanning the totality of the synaptic cleft as defined by WGA staining. The final acquisition settings included a sequential acquisition in the order 647/594/488, a frame size of 528 × 528 px, a pixel time of 7.95 μs, a pixel size of $0.043 \times 0.043 \times 0.16$ μm, and a doubled pixel sampling with bidirectional mean intensity averaging of acquisition lines. Time-lapse imaging was performed using ×20 magnification and the super-resolution SR4Y mode and frames were acquired every 30 s. Analyses were performed using the ZEN 3.2 sys-tem blue edition (Carl Zeiss Microscopy GmbH) and Fiji v2.1.0/1.53c (build 5f23140693)[73].

**eTIRF-SIM**. A custom-built eTIRF-SIM microscope system was used and detailed elsewhere[74]. Structured illumination was obtained via a grating pattern generated

by a ferroelectric spatial light modulator (SLM, Forth Dimension Displays, QXGA3DM). After, the first diffraction orders are selected by the mask and sent to the Olympus IX83 microscope head. The distance between diffraction orders were tuned by SLM settings ultimately defining the illumination angle and therefore the TIRF depth. Excitation wavelengths of 488 nm, 560 nm, and 640 nm were used (MPB communications Inc., 500 mW, 2RU-VFL-P-500-488-B1R, 2RU-VFL-P-500-560-B1R). Sample illumination was carried out using a high-NA TIRF objective (Olympus Plan-Apochromat 100 × 1.49 NA). The emitted fluorescence was collected by the same objective and sent onto sCMOS cameras recording the raw data (Hamamatsu, Orca Flash 4.0 v2 sCMOS). The excitation numerical aperture (NA) was adjusted for each wavelength by changing the period of the grating pattern at the SLM, which allows controlling the TIRF angle and, therefore, the penetration depth of the evanescent wave. To achieve TIRF-SIM illumination at the interface between the cells and SLB, both excitation lights were sent with an incident NA ranging from 1.38 to 1.41. Prior to experiment acquisition alignment was perform by imaging 100 nm Tetraspeck fluorescent beads in all three excitation colors. Then, using Fiji plugin MultiStackReg – the beads images were used to adjust images and compensate for chromatic aberrations. A total of 9 raw images were acquired per frame and for a single excitation wavelength before switching to the next wavelength. Then, raw images were processed and reconstructed into SIM images by custom made software or ImageJ fairSIM plugin. All experiments were performed at physiological conditions using a micro-incubator (H301, Okolabs, Italy) at 37 °C and 5% $CO_2$. For each frame, we used an acquisition time between 20 and 300 ms depending on the fluorescence signal levels and 2 colors, 18 frames total) every 0.4–5 s. One color super-resolved image was reconstructed from 9 raw image frames (3 angles and 3 phases) using a reconstruction method described previously[75,76].

**CRISPR/Cas9-editing of primary human T cells**. The ablation of genes of interests was achieved by transfection with pre-assembled Cas9 RNP complexes. Prior to transfection, and after 48 h of stimulation with Human T-activator CD3/CD28 Dynabeads, $CD4^+$ lymphoblasts were collected for removal of stimulating beads and then washed three times with 10 volumes of prewarmed OptiMEM I medium (ThermoFisher Scientific, #31985070). Cells were resuspended to a final $3 × 10^7$ cells/mL such that $1.5 × 10^6$ cells/mL were contained in 50 µL of cell suspension. In parallel, RNP complexes were assembled in vitro in two steps. First, 150 pmol of Alt-R CRISPR-Cas9 tracrRNA (200 µM stock; Integrated DNA Technologies (IDT), #1072534) were mixed with 150 pmol of Alt-R CRISPR-Cas9 predesigned crRNA (200 µM stock) and then incubated at 95 °C for 5 min and the resultant duplex guide RNA allowed to cool to room temperature. All Alt-R CRISPR-Cas9 crRNA sequences and cat# are given in Supplementary Table 2 (all on- and off-target scores were optimized by IDT). The duplex gRNA was then mixed with 150 pmol of Alt-R *S. pyogenes* CRISPR-Cas9 Nuclease V3 (IDT, #1081061, 20 µM stock) and incubated at 37 °C for 15 min. The resultant preformed RNPs were allowed to cool to room temperature and then supplemented with 150 pmol of Alt-R Cas9 Electroporation Enhancer (IDT, #1075916, 200 µM stock). Cells were then mixed with the RNP solution and immediately transferred to a 2-mm cuvette (Bio-Rad) and electroporated at 300 V for 2 ms using an ECM830 Square Wave electroporator. Immediately after transfection, cells were recovered with prewarmed, IL-2 supplemented RPMI 1640 media and expanded for another 6 days. Synaptic transfer experiments to BSLB or planar SLB were performed on day 7 or 8 of culture and protein expression controls were carried out in parallel.

**Western blotting**. Whole-cell lysates (WCL) were prepared by resuspending cell pellets in RIPA lysis and extraction buffer (ThermoFisher Scientific, #89901) containing a Protease/Phosphatase inhibitor cocktail (Cell Signaling Technology (CST); #5872) to a final concentration of $2 × 10^7$ cells/mL. After sonication at +4 °C (10 cycles of 30 s on/30 s off), WCL were centrifuged at 10,000 × g for 10 min at +4 °C, and the supernatants collected, mixed with loading solution and denatured at +95 °C for 10 min. For immunoblotting of BSLB eluted material, after centrifugation at 120,000× g for 4 h at +4 °C, pellets were resuspended in RIPA lysis buffer containing a Protease/Phosphatase inhibitor cocktail, mixed with loading solution and denatured at +95 °C for 10 min. When indicated, lysed eluates and EVs from an equivalent number of originating cells were resolved to compare different vesicle populations. As positive control we used WCL equivalent to $2.25 × 10^5$ $CD4^+$ lymphoblasts. Similarly, for CRISPR/Cas9-edited cells, WCL equivalent to $3 × 10^5$ cells were used per lane. Samples were resolved using 4–15% Mini-PROTEAN SDS-PAGE gel (Bio-Rad; #4561084), transferred to 0.45 µm nitrocellulose membranes (Bio-Rad, #1620115), blocked with PBS containing 5% BSA and incubated with the following primary antibodies: rabbit anti-human CD40 Ligand clone D5J9Y (CST, #15094), mouse anti-β-actin clone 8H10D10 (CST, #3700), mouse anti-CD81 clone M38 (Invitrogen, #10630D), rabbit anti-ALIX clone EPR15314 (#ab186429), and rabbit anti-TSG101 clone EPR7130B (#ab125011). Then, membranes were incubated with IRDye® 680RD donkey anti-mouse IgG (H+L; LI-COR, #926-68072) and IRDye® 800CW donkey anti-rabbit IgG (H+L; LI-COR, #925-32213) secondary antibodies following manufacturer guidelines. After four washes, membranes were imaged and analyzed using the Odyssey® CLx Near-Infrared detection system equipped with the Image Studio™

Lite quantification software (LI-COR, Lincoln, NE). Uncropped, unprocessed and annotated immunoblot images have been included as part of the Source Data.

**RNA sequencing and miR analyses**. Total cell RNA was extracted using the miRNeasy Tissue/Cells Advanced Mini Kit (Qiagen, #217604). Purified RNA yields and quality were assessed via Agilent 2100 Bioanalyzer using the Agilent 6000 RNA Pico chips (Agilent Technologies, # 5067-1513). Before library preparation, the material was further quantified using RiboGreen (Invitrogen) on the FLUOstar OPTIMA plate reader (BMG Labtech) and the size profile and integrity analyzed on the 2200 or 4200 TapeStation (Agilent, RNA ScreenTape). Input material was normalized to 200 ng or maximum mass for input volume prior to library preparation. Small RNA library preparation was completed using NEBNext Small RNA kit (NEB) following the manufacturer's instructions and applying the low input protocol modifications. Libraries were amplified (15 cycles) on a Tetrad (Bio-Rad) using in-house unique dual indexing primers[77]. Size selection was performed using Pippin Prep instrument (Sage Science) using the 3% Agarose, dye-free gel with internal standards (size selection: 120 to 230 bp). Individual libraries were normalized using Qubit, and the size profile was analyzed on the 2200 or 4200 TapeStation. 10 nM libraries were denatured and further diluted prior to loading on the sequencer. Two runs of single-end sequencing were performed; run one was performed in an Illumina HiSeq 2500 system (1 × 60) and run two was performed using Illumina NextSeq 500/550 v2.5 Kits (75 cycles). Quality control and processing of the raw sequencing data were performed using sRNAbench[78] and miRQC[79] and sRNAtoolboxDB, which allowed the assessment of sequencing yield, quality, percentage of miR-mapping reads, read length distribution and relative abundance of fragments from other RNA species. Functional and biological pathway enrichment analyses were performed on annotated miR species shared between the two independent sequencing runs and enriched in each EV category (i.e., tSV and EVs). MIENTURNET and the KEGG annotation database were used. Statistical analysis for functional and biological pathway enrichment analyses using MIENTURNET was carried out by calculating the adjusted P value resulting from the hypergeometric statistic test (false discovery rate (FDR), $Q = 0.05$). Motif analyses of bulk, enriched miR sequences in each sample were performed using MEME (Multiple Em for Motif Elucidation 5.1.0)[53]. To dissect the miR-target gene interactome of our enriched miR we used miRNet 2.0[80] by first mapping input miR to the miR interaction knowledgebase comprising annotations from miRbase, miTarBase, and ExoCarta together with interactions with other miR, genes, and transcription factors from TransmiR 2.0, ENCODE, JASPAR, and ChEA. The output of the 'miRs' module of miRNet 2.0 provided the motif miR-target gene interactome enrichment analysis.

**Mass spectrometry**. Samples isolated by differential centrifugation were concentrated using 100,000 NMWL centrifugal filters (Amicon® Ultra, #UFC510024, Merck Millipore Ltd.) and prepared in S-Trap™ spin columns (#C02-micro-80) following manufacturer recommendations. Briefly, samples were reduced with 5 µL of 10 mM TCEP and alkylated with 50 mM of IAA for 30 min each, then acidified with 12% phosphoric acid 10:1 vol:vol, and transferred to S-trap columns. Then, samples were precipitated using 1:8 vol:vol dilution of each sample in 90% methanol in 100 mM TEAB. Samples were then washed three times with 90% methanol in 100 mM TEAB. Samples were then resuspended in 50 µL of 50 mM TEAB and digested with trypsin (Promega, #V1115) overnight at 37 °C. Peptides were eluted from the S-Trap by spinning for 1 min at 1500 × g with 80 µL of 50 mM ammonium bicarbonate, 80 µL 0.1% FA and finally 80 µL of 50% ACN 0.1% FA. The eluates were dried down in a vacuum centrifuge and resuspended in 2% ACN 0.1% TFA prior to off-line high-pH reversed-phase fractionation using RP-S cartridges pre-primed with 100 µL ACN at 300 µL/min and equilibrated with 50 µL of 2% ACN 0.1% TFA at 10 µL/min. Samples were loaded at 5 mL/min and divided into eight fractions (elution steps), which were run individually. Elution was performed with increasing concentrations of 90% ACN, pH 10 in water, including final 5%, 10%, 12.5%, 15%, 20%, 22.5%, 25 and 50%. Fractions were further dried down in a vacuum centrifuge and resuspended in loading buffer. For LC-MS/MS acquisition 50–80 ng peptides were loaded onto preconditioned Evotips containing 0.1% FA in water. Preconditioning was performed by pre-priming of isopropanol-soaked tips with 20 µL of ACN 0.1% FA, following by centrifugation for 1 min at 700 × g, equilibration in water 0.1% FA and a final centrifugation for 1 min at 700 × g). Samples were run on a LC-MS system comprised of an Evosep One and Bruker timsTOF Pro. Peptides were separated on an 8 cm analytical C18 column (Evosep, 3 µm beads, 100 µm ID) using the pre-set 60 samples per day gradient on the Evosep One. Acquisition was done in PASEF mode (four PASEF frames, three cycles overlap, oTOF control v6.0.0.12) including an ion mobility window between $1/k0$ start = 0.85 Vs/cm² to $1/k0$ end = 1.3 Vs/cm², a ramp time of 100 ms with locked duty cycle and a mass range of 100–1700 m/z. For proteomic analyses the raw files were searched against the reviewed Uniprot *Homo sapiens* database (retrieved 2,01,80,131) using MaxQuant version 1.6.10.43[81] and its built-in contaminant database using tryptic specificity and allowing two missed cleavages. Gene function and biological pathway analysis of identified proteins was performed using PANTHER[48] following Fisher's exact test and the Benjamini-Hochberg FDR correction using default Q values. The result of GO enrichment analyses was further filtered using FunSet[82] with the Benjamini-Hochberg correction (FDR with $Q = 0.05$).

**Statistical analyses**. Normality tests were performed using either Shapiro–Wilk or Kolmogorov–Smirnov tests. Statistical significance was determined by multiple comparisons performed either by one-way analysis of variance, or multiple two-tailed $t$ tests. Non-linear regressions using three or four parameters $F$ tests using least squares regression as a fitting method, with no weighting, and 95% confidence interval levels were used when indicated. All statistical tests were performed in GraphPad Prism v8 and are detailed in each figure legend. Means of significance are detailed in each figure legend.

**Reporting summary**. Further information on research design is available in the Nature Research Reporting Summary linked to this article.

## Data availability

The mass spectrometry proteomics data generated in this study have been deposited to the ProteomeXchange Consortium via the PRIDE[83] partner repository with the dataset identifier PXD033917. The RNA sequencing data generated in this study have been deposited in the Gene Expression Omnibus (GEO)[84,85] under the accession number GSE181216. Plasmids encoding recombinant proteins (12-His) and anti-CD3ε-Fab are available upon request to the corresponding authors. Source data are provided in this paper.

## Code availability

All bioinformatics analyses were performed with previously published tools and are referenced in the Methods section. The source codes for RNA sequencing and proteomic analyses were extracted from GitHub; miRNA-seq quality control (https://github.com/sert23/miRNAQC), gene set enrichment analyses with funset (https://github.com/mlhale/funset-enrichment-visualization) and ShinyGo (https://github.com/iDEP-SDSU/idep/tree/master/shinyapps/go61).

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

## Acknowledgements

We are grateful to our laboratory members and the Kennedy Institute of Rheumatology community for constructive scientific discussions, especially to James Felce, David Depoil, Jonathan Webber, Štefan Balint, Alexander Mørch, and Kristina Correa. We thank the technical support of Heather Rada, Kellie Johnson, and Ekaterina Zvezdova (the latter two from BioLegend). We thank Professor Catarina E. Hioe for kindly providing the HIV-1 gp120 protein. We would also like to thank all the anonymized blood donors who contributed to our study. This work was funded by Wellcome Trust Principal Research Fellowship 100262Z/12/Z, the ERC Advanced Grant (SYNECT AdG 670930), and the Kennedy Trust for Rheumatology Research (KTRR) (all three to M.L.D.). P.F.C.D was supported by EMBO Long-Term Fellowship (ALTF 1420–2015, in conjunction with the European Commission (LTFCOFUND2013, GA-2013-609409) and Marie Sklodowska-Curie Actions) and Oxford-Bristol Myers Squibb Fellowship. A.K. was supported by H2020 and the Research Council of Norway (in conjunction with Marie Sklodowska-Curie Actions 275466; to A.K.). M.F. and H.C.Y. thank the Wellcome Trust (212343/Z/18/Z) and EPSRC (EP/S004459/1). The eTIRF-SIM platform was built-in collaboration with Micron (www.micronoxford.com), an Oxford-wide advanced microscopy technology consortium supported by Wellcome Strategic Awards (091911 and 107457), and with additional funds from an MRC/EPSRC/BBSRC next-generation imaging award and the Kennedy Trust for Rheumatology Research through the Kennedy Institute Cell Dynamics Platform. We acknowledge the generous support of the Kennedy Trust for Rheumatology Research, IDRM, and Carl Zeiss GMBH for the Airyscan LSM 980 confocal microscope used in this research. Y.P., T.D., and R.F. were supported by the Chinese Academy of Medical Sciences (CAMS) Innovation Fund for Medical Sciences (CIFMS), China (grant number: 2018-I2M-2-002) and UK Medical Research Council (MRC); E.S. was supported by Newton-Katip Celebi Institutional Links grant (352333122) and SciLifeLab fellowship (to E.S.). F.S-M. was supported by grants SAF2017-82886-R from the Spanish Ministry of Economy and Competitiveness (MINECO), and "La Caixa" Banking Foundation (HR17-00016). We thank the NIH Tetramer Core Facility for the synthesis of the HLA-DRB1*09:01 monomers used in this study. We thank the Oxford Genomics Centre at the Wellcome Centre for Human Genetics (funded by Wellcome Trust grant reference 203141/Z/16/Z) for the generation and initial processing of the sequencing data. Finally, we thank the MS laboratory at the Target Discovery Institute NDM (Oxford) led by Benedikt M. Kessler. Pablo F. Céspedes is also known as Pablo F. Céspedes-Donoso (https://orcid.org/0000-0002-1641-4107).

## Author contributions

P.F.C. wrote the manuscript, secured funding, and conceived, designed, executed, analyzed, and interpreted data. A.K.J. executed and interpreted transcriptomics and

proteomics bioinformatics analyses. L.F-M., F.S-M., S.E., and M.A. executed sncRNA sequencing in two independent facilities. D.G.S. performed fluorescence-activated cell sorting of BSLBs for proteomic and sncRNA sequencing. S.V., A.K., L.C., E.K., and C.G. provided essential infrastructure and technical support; H.C-Y and M.F. performed TIRF-SIM experiments; Y.P. and T.D. provided T-cell clones and related infrastructure, E.J. provided electron microscopy support, J.A.S-F and O.D. provided CART cells and related infrastructure, E.S. performed FCS on BSLBs; B.P., A.L., and D.A. performed blinded and independent NanoFCM measurements; S.H. and R.F. performed LC-MS/MS, and M.L.D. wrote the manuscript, secured funding, and interpreted the experiments. All authors read the manuscript and provided critical suggestions shaping the submitted and revised versions of the manuscript.

## Competing interests

Ben Peacock, Alice Law, and Dimitri Aubert are employed by NanoFCM Co., Ltd. The remaining authors declare no competing interests.
