## [Peer Review File · Nature Communications]

T cell trans-synaptic vesicles are distinct and carry greater effector content than constitutive extracellular vesiclesREVIEWER COMMENTS

Reviewer #1 (Extracellular vesicle) (Remarks to the Author):

The manuscript by Cespedes PF, Jainarayanan A et al., describes a new use for bead-supported lipid bilayers (BSLB) in the capture and study of trans-synaptic vesicles (tSV) and synaptic extracellular vesicles (sEV) in the immunological synapsis. The Authors provide convincing data in support of this technology to isolate tSV and sEV from T helper, T cytotoxic and Treg cells, and show that this technology is successful in characterizing the different properties and cargo content of the two types of EVs.

This study will certainly help further applying this technology to isolate sub-populations of EVs in the immunological synapsis and mechanistically determine the contribution of specific cargo molecules to the biology of T cell-APC interaction.

The only minor concerns I have are that the graphic abstract is a little confusing and should be made more straightforward and I identified a typo on line 81 (should be "have been studied" instead of "have been studies"). Moreover, in the "Method" section for the CRISPR/Cas9 experiment, it is not described what was the source of the tracer RNA and crRNAs used for these experiments.

Other than this, I applaud the rigor of the Authors in conducting these experiments and the provided experimental details that make the experiments totally reproducible.

Reviewer #2 (Lipid bilayer biology) (Remarks to the Author):

The manuscript by Céspedes and colleagues presents a novel approach based on BSLBs used as synthetic antigen-presenting cells to "physically" capture (and subsequently isolate and examine) vesicles that are secreted by the T cells at the immunological synapse. Besides introducing this appealing method, the authors also aim at investigating what they call the fourth signal at the IS: the trans-synaptic vesicles.

Let me start by saying that this work is very interesting, scientifically sound, well-executed and thought-provoking. As such it merits consideration by NCOMMS. It is somewhat 'descriptive' but this is not a limitation in this case as the study addresses a long-standing challenge and the approach can be applied to other cell-cell interaction systems.

I do have however a couple of concerns and questions that I hope the authors are willing to address.

My main concerns revolve around the following two aspects:

- 1) This manuscript -despite being written in perfect English- is very difficult to read. One reason I identify is the monumental amount of data that are included (the amount of experiments and controls and quantification is mesmerizing), the overwhelming amount of protein names and abbreviations. The other reason is that it does require quite a substantial immunological background to be able to appreciate the information provided. While this obviously shows how scientifically accurate this study has been carried out, it does make it very hard to follow and to make sense of the data. There are three stories hidden in this manuscript: i) the method itself

with its applicability, power and standardization (Fig 1 plus three supplementary figures for a total of 37! Panels); ii) the different tSV made by the three different T cell types (Fig 2-4 plus related supplementary figures) and iii) the RNA content of these tSV (Fig 5 and the related suppl fig).

2) The duration of IS between DC and T cells has been reported to be up to hours. The CTLs make multiple hits (synapses) with target cells before killing. Where should we place the tSV isolated here (result of mostly 90 min T cell-BSLB interaction time) in the normal temporal scale of a synapse formation? If vesicle composition/amount changes over time and maybe also as a function of the various DC subsets that stimulate T cells, how does this approach can help us revealing this? The possibility to isolate and further characterize these tSV by this method is really very appealing but where is the evidence they really resemble the tSV at the real IS? I would be ashamed to ask for additional work with real DC-T cell synapses, but maybe the authors should comment on this. The discussion is a good attempt to make sense of all the data but it has a high degree of speculation.

Additional points:

3) Fig 2: Most of the Plots F-N show ad hoc only two out of the three T cell types (Treg, TH and CTL), with no clear reason provided to justify this choice. Nat Comm readership is pretty interdisciplinary, so better explanation can help understanding the choice.

4) Fig 3: panel E and F are the same panel, I guess F should show the T cell types.

5) Characterization of the protein contents (fig 1-4) provides an enormous amount of information but remains a bit in the air: would an overview as done for the RNA in Fig. 5A-C be helpful here too?

6) Is there any connection/correlation to be made between the tSV content in Fig 1-4 and the content in Fig 5? The two parts are disconnected.

7) Fig 5: At what anti-CD3e-Fab density was performed fig 5? Considering the differences in vesicle content as a function of anti-CD3 density, I am wondering whether this is also true for the RNA and RBP content. Also, the authors state this figure shows one representative donor out of 8... but I am concerned about the variability among the different donors. This information should somehow be provided.

Reviewer #3 (T cell activation, immune synapse) (Remarks to the Author):

Review – Cespedes, et al. Synthetic antigen-presenting cells reveal the diversity and functional specialisation of extracellular vesicles composing the fourth signal of T cell immunological synapses.

The manuscript describes an adapted method, previously described in eLife (2019), to manipulate membrane proteins and RNAs captured from cells interacting with bead supported

lipid bilayers (BSLBs). The approach allows comparison between extracellular vesicles (EVs) and trans-synaptic vesicles (tSVs), highlighting the heterogeneity of secreted signals dependent on the context of ligands presented on the BSLBs. The authors strengthen their claim for a fourth signal in T cell activation and here provide the tools for a multiparametric comparison. I found the work compelling and the applications will be of great interest to the field.

After a helpful description of the methods and analyses used to interrogate tSVs and EVs the authors walk us through at least four main applications of BSLBs in characterizing synaptic release of biomolecules from T cells. First, synaptic transfer was related to different T cell subsets, including TH, Treg, CTLs, and some additional work on CAR-Ts. Next, an in-depth analysis of CD40L+ tSVs and highlighted the heterogeneity of tSV release and what molecules correlate with this transfer. This analysis was also applied to different clones of TCRs recognizing the same antigenic peptide (ie. signal one). Likewise, different costimulatory receptors were tested in various combinations to probe the contribution of signal two influencing signal 4. Next, CRISPR editing was used to assay how critical pathways in vesicular trafficking and other endogenous elements influence tSVs on BSLBs. Lastly, miRNAs and RNA binding proteins are profiled in tSVs. I found this last piece most intriguing and the authors provide just a taste of the possible applications of BSLBs in characterizing T cells breadth of signaling components. I would prefer the last bit to cover RNA binding proteins discovered from the proteomics analysis first and then to dive into the miRNAs as a subspecies of RNAs with potential for target cell influence. I believe this will help the flow of the paper. Besides a few editorial issues I believe this paper to be of great interest and worthy of publication.

Editorial issues:

There is some confusion between Fig. S3 legend and text (line 163) for the time of drug treatment.

Figure 3E is likely wrong since the text refers to something different and panel F is the same figure.

There is a missing figure header for Fig. S7E

How was RNA visualized? This should be explained in the text.

REVIEWER COMMENTS

Reviewer #1 (Extracellular vesicle) (Remarks to the Author):

The manuscript by Cespedes PF, Jainarayanan A et al., describes a new use for bead-supported lipid bilayers (BSLB) in the capture and study of trans-synaptic vesicles (tSV) and synaptic extracellular vesicles (sEV) in the immunological synapsis. The Authors provide convincing data in support of this technology to isolate tSV and sEV from T helper, T cytotoxic and Treg cells, and show that this technology is successful in characterising the different properties and cargo content of the two types of EVs.

This study will certainly help further applying this technology to isolate sub-populations of EVs in the immunological synapsis and mechanistically determine the contribution of specific cargo molecules to the biology of T cell-APC interaction.

Comment 1: The only minor concerns I have are that the graphic abstract is a little confusing and should be made more straightforward

Answer: We had to remove the graphical abstract to comply with Nature Communications formatting guidelines.

Comment 2: I identified a typo on line 81 (should be “have been studied” instead of “have been studies”).

Answer: We thank the reviewer to pointing out to this error. We have corrected the manuscript accordingly.

Comment 3: Moreover, in the “Method” section for the CRISPR/Cas9 experiment, it is not described what was the source of the tracer RNA and crRNAs used for these experiments.

Answer: We took advantage of the no space limits of Supplementary Information to provide full details on the materials and methods related to the genome-editing using transfection of preformed CRISPR/Cas9 ribonucleoprotein complexes. Please refer to Pages 31-32 and lines 546-567 of the Supplementary Information and Supplementary Data 6.

Comment 4: Other than this, I applaud the rigor of the Authors in conducting these experiments and the provided experimental details that make the experiments totally reproducible.

Answer: We appreciate the positive comments of the reviewer and we hope the revised version of the manuscript is acceptable for publication in Nature Communications.

Reviewer #2 (Lipid bilayer biology) (Remarks to the Author):

The manuscript by Céspedes and colleagues presents a novel approach based on BSLBs used as synthetic antigen-presenting cells to "physically" capture (and subsequently isolate and examine) vesicles that are secreted by the T cells at the immunological synapse. Besides introducing this appealing method, the authors also aim at investigating what they call the fourth signal at the IS: the trans-synaptic vesicles.

Let me start by saying that this work is very interesting, scientifically sound, well-executed and thought-provoking. As such it merits consideration by NCOMMS. It is somewhat 'descriptive' but this is not a limitation in this case as the study addresses a long-standing challenge and the approach can be applied to other cell-cell interaction systems.

I do have however a couple of concerns and questions that I hope the authors are willing to address.

My main concerns revolve around the following two aspects:

Comment 1: This manuscript -despite being written in perfect English- is very difficult to read. One reason I identify is the monumental amount of data that are included (the amount of experiments and controls and quantification is mesmerising), the overwhelming amount of protein names and abbreviations. The other reason is that it does require quite a substantial immunological background to be able to appreciate the information provided. While this obviously shows how scientifically accurate this study has been carried out, it does make it very hard to follow and to make sense of the data. There are three stories hidden in this manuscript: i) the method itself with its applicability, power and standardisation (Fig 1 plus three supplementary figures for a total of 37! Panels); ii) the different tSV made by the three different T cell types (Fig 2-4 plus related supplementary figures) and iii) the RNA content of these tSV (Fig 5 and the related suppl fig).

Answer: We appreciate the reviewer's positive comments, and we are even more grateful for his/her criticisms of the format and presentation of the data. We have taken some time to rethink and rewrite some parts of the manuscript to connect better the different sections, and introduced changes to help readers navigate the concepts and data. We summarise the edits below:

- 1) We updated the title and abstract so there is a better introduction to the major sections mentioned by the reviewer.
- 2) Whenever possible and to facilitate reading, we have reduced the number of abbreviations used.
- 3) At the end of the Supplementary Discussion, we also included a List of Abbreviations to help readers navigate the data (Please refer to Pages 40-41, lines 755-781).
- 4) We reduced the extent of the manuscript to meet the formatting requirements of Nature Communications (maximum of 6,000 words without Figure Legends). We reduced the Discussion's extent and focused the attention on the questions raised by the reviewers.

- 5) We have discussed other relevant aspects of tSV biology in the Supplementary Discussion.
- 6) We decided to follow the recommendations of reviewer three, who also suggested changes in the order of presentation of the result; in the last section, we presented the proteomics data first and then the sncRNA sequencing data (Please refer to new Figure 5 and the new Supplementary Figures 6 and 7).

Comment 2: The duration of IS between DC and T cells has been reported to be up to hours. The CTLs make multiple hits (synapses) with target cells before killing. Where should we place the tSV isolated here (result of mostly 90 min T cell-BSLB interaction time) in the normal temporal scale of a synapse formation? If vesicle composition/amount changes over time and maybe also as a function of the various DC subsets that stimulate T cells, how does this approach can help us revealing this?

Answer: We have performed experiments to address how the hours-long interaction of T cells and BSLB enables the identification of tSV released in prolonged contacts. We demonstrate the transfer of the T cell membrane and related tSV markers to BSLBs that mimic the composition of a DC membrane. In new **Supplementary Figures 8a and 8b**, we show that following 24h of co-culture with TH cells, BSLB efficiently capture tSV, as evidenced by the tracking of several vesicle markers. In **Supplementary Figure 8c**, we also compare the quantitative differences observed between 90 and 24h of T cell-BSLB co-culturing. In the Discussion and Supplementary Information, we contrast these results with the measurements performed on DCs following 24 h co-culturing with TH. **Please refer to Discussion Page 20-21 and lines 479-500, and Supplementary Information pages 17-18 and lines 197-230.**

In the Introduction section, we have also discussed the kinetics of interaction between T cells and APCs and evidence demonstrating the transfer of tSV markers within minutes¹⁻³ of interaction. Please refer to pages 3-4, lines 71-85.

Comment 3: The possibility to isolate and further characterise these tSV by this method is really very appealing but where is the evidence they really resemble the tSV at the real IS? I would be ashamed to ask for additional work with real DC-T cell synapses, but maybe the authors should comment on this. The Discussion is a good attempt to make sense of all the data but it has a high degree of speculation.

Answer: We have performed complementary experiments to demonstrate the similarities between vesicles captured by BSLBs and those secreted within the synapse of TH cells and DCs. In our previous work, we have reconstituted synthetic tSV containing physiological densities of CD40L and demonstrated that these sufficed to induce activation of monocyte-derived DCs as evidenced by ICAM1 and CD86 upregulation⁴. Similarly, we have previously reported activating human DCs⁴ and mouse B cells³ on patches of TCR⁺ vesicles left on planar supported lipid bilayers. We have also shown by microscopy that mouse and human T cells transfer TCR⁺ puncta to B cells driving their activation and, more recently, to CHO cells expressing class-II MHC antigenic complexes¹. If tSV deposited in planar SLB and BSLBs are identical to those released in cell-cell conjugates, we hypothesised that a similar activation signature would be observed in moDCs co-cultured with TH cells. In new **Supplementary Figure 8 d-k**, we show that moDCs engulf T cell membrane in hours-long co-cultures regardless of their antigenicity. However, antigenic DCs (i.e., SEB+) showed a significantly superior internalisation of T cell membrane and upregulation of ICAM1, CD86 and CD40 and acquisition of CD40L, consistent with our prediction.

References:

- 1 Audun Kvalvaag, P. F. C., Salvatore Valvo, David G Saliba, Elke Kurz, Kseniya Korobchevskaya, Michael L Dustin. Clathrin mediates both internalisation and vesicular release of triggered T cell receptor at the immunological synapse through distinct adaptors. *bioRxiv*, doi:<https://doi.org/10.1101/2022.02.02.478780> (2022).
- 2 Balint, S. *et al.* Supramolecular attack particles are autonomous killing entities released from cytotoxic T cells. *Science* **368**, 897-901, doi:10.1126/science.aay9207 (2020).
- 3 Choudhuri, K. *et al.* Polarised release of T-cell-receptor-enriched microvesicles at the immunological synapse. *Nature* **507**, 118-123, doi:10.1038/nature12951 (2014).
- 4 Saliba, D. G. *et al.* Composition and structure of synaptic ectosomes exporting antigen receptor linked to functional CD40 ligand from helper T cells. *Elife* **8**, doi:10.7554/eLife.47528 (2019).

Additional points:

Comment 4: Fig 2: Most of the Plots F-N show ad hoc only two out of the three T cell types (Treg, TH and CTL), with no clear reason provided to justify this choice. Nat Comm readership is pretty interdisciplinary, so better explanation can help understanding the choice.

Answer: We have explained this in the manuscript's Results section. Please, refer to page 9, lines 211-213.

Comment 5: Fig 3: panel E and F are the same panel, I guess F should show the T cell types.

Answer: We apologise for this error. We have corrected the manuscript accordingly.

Comment 6: Characterisation of the protein contents (fig 1-4) provides an enormous amount of information but remains a bit in the air: would an overview as done for the RNA in Fig. 5A-C be helpful here too?

Answer: We have corrected the manuscript accordingly to present first a summary of the proteomics data, followed by the presentation of our small RNA sequencing data (Please refer to **new Figure 5**).

Comment 7: Is there any connection/correlation to be made between the tSV content in Fig 1-4 and the content in Fig 5? The two parts are disconnected.

Answer: We have performed several edits to the manuscript to connect better the presentation of the results in Fig 1-4 to those of Fig. 5:

1. In the introduction, we mentioned the different particles identified by the literature and the different effectors in more detail, highlighting microRNAs as relevant trans-cellular communication entities and the importance of their study with BSLBs.
2. In the results section, we have introduced the concept of low copy numbers of miR being transferred via EVs and the potential role of the immune synapse as a hub favouring their acute transfer (please refer to page 16, lines 374-383, and page 18, lines 424-428).
3. For new figure 5, we have taken the suggestions made by reviewer #3 and presented the proteomics data first, followed by the presentation of RNA sequencing data.

Comment 8: Fig 5: At what anti-CD3e-Fab density was performed fig 5? Considering the differences in vesicle content as a function of anti-CD3 density, I am wondering whether this is also true for the RNA and RBP content.

Answer: We thank the reviewer for pointing out the missing information in this section. The density of anti-CD3 Fab used (250 molec./ μm^2) was determined empirically as the one sufficient to induce a significant transfer of TCR such that all sorted BSLBs were TCR⁺ (Fig. S7a). We compare the number of miR species transferred in response to adhesion (null BSLBs) and those transferred in response to bilayers presenting 250 molec./ μm^2 of anti-CD3 Fab (please refer to **new Supplementary Figure 7c**).

We agree that understanding whether RBP and RNAs change across different densities of signal one and composition of signal two is of biological and translational interest. However, this is currently beyond the scope of this work. The significant queue in sequencing projects (due to COVID-19 and Brexit) generated a significant delay, and by the time of this submission, we could not generate data to answer this question. We are working on securing further funding and unravelling sorting mechanisms linking signal integration with the composition of RNA-editing machinery transferred across diverse cell-cell interfaces.

Comment 9: Also, the authors state this figure shows one representative donor out of 8... but I am concerned about the variability among the different donors. This information should somehow be provided.

Answer: We have amended the manuscript accordingly to clarify that the data shown in current Figure 5 integrates the information from all eight donors. We have also provided deposited the raw data in GEO and Source Data as part of this submission. With this, we hope the information will be openly available to colleagues aiming at contrasting these analyses with those of new datasets in the future.

Reviewer #3 (T cell activation, immune synapse) (Remarks to the Author):

Review – Cespedes, et al. Synthetic antigen-presenting cells reveal the diversity and functional specialisation of extracellular vesicles composing the fourth signal of T cell immunological synapses.

The manuscript describes an adapted method, previously described in eLife (2019), to manipulate membrane proteins and RNAs captured from cells interacting with bead supported lipid bilayers (BSLBs). The approach allows comparison between extracellular vesicles (EVs) and trans-synaptic vesicles (tSVs), highlighting the heterogeneity of secreted signals dependent on the context of ligands presented on the BSLBs. The authors strengthen their claim for a fourth signal in T cell activation and here provide the tools for a multiparametric comparison. I found the work compelling and the applications will be of great interest to the field.

After a helpful description of the methods and analyses used to interrogate tSVs and EVs the authors walk us through at least four main applications of BSLBs in characterising synaptic release of biomolecules from T cells. First, synaptic transfer was related to different T cell subsets, including TH, Treg, CTLs, and some additional work on CAR-Ts. Next, an in-depth analysis of CD40L+ tSVs and highlighted the heterogeneity of tSV release and what molecules correlate with this transfer. This analysis was also applied to different clones of TCRs recognising the same antigenic peptide (ie. signal one). Likewise, different costimulatory receptors were tested in various combinations to probe the contribution of signal two influencing signal 4. Next, CRISPR editing was used to assay how critical pathways in vesicular trafficking and other endogenous elements influence tSVs on BSLBs. Lastly, miRNAs and RNA binding proteins are profiled in tSVs. I found this last piece most intriguing

and the authors provide just a taste of the possible applications of BSLBs in characterising T cells breadth of signalling components.

I would prefer the last bit to cover RNA binding proteins discovered from the proteomics analysis first and then to dive into the miRNAs as a subspecies of RNAs with potential for target cell influence. I believe this will help the flow of the paper. Besides a few editorial issues I believe this paper to be of great interest and worthy of publication.

Answer: We appreciate the positive comments of the reviewer. We have corrected the manuscript accordingly and hope its revised version is acceptable for publication in Nature Communications.

Editorial issues:

There is some confusion between Fig. S3 legend and text (line 163) for the time of drug treatment.

Answer: We apologise for this error. We have corrected the manuscript and figure legend to make this clearer to readers. Please refer to the Result section, page 7 and lines 168-171, and the Supplementary Information pages 6-7 and lines 64-68.

Figure 3E is likely wrong since the text refers to something different and panel F is the same figure.

Answer: We apologise for this error. We have corrected the figure accordingly.

There is a missing figure header for Fig. S7E

Answer: We apologise for this error. We have corrected the figure accordingly (Please refer to **the new Supplementary Fig. 6e**).

How was RNA visualised? This should be explained in the text.

Answer: We have provided complete details of the material and methods for the labelling and live tracking of RNA in T cell synapses. Please refer to Supplementary Material and Methods, pages 27-28, and lines 444-456.

REVIEWERS' COMMENTS

Reviewer #2 (Remarks to the Author):

The authors have addressed all my concerns and properly revised the manuscript. I fully support publication of this study.

Reviewer #3 (Remarks to the Author):

I'm satisfied with the author's responses to my queries.